



# Anomalous summertime CO₂ sink in the subpolar Southern Ocean promoted by early 2021 sea ice retreat

Kirtana Naëck[1*], Jacqueline Boutin[1], Sebastiaan Swart[2], Marcel du Plessis[2], Liliane Merlivat[1], Laurence Beaumont[3], Antonio Lourenco[1], Francesco d'Ovidio[1], Louise Rousselet[1], Brian Ward[4], Jean-Baptiste Sallée[1]

[1] Sorbonne Université, CNRS, IRD, MNHN, Laboratoire d'Océanographie et du Climat : Expérimentations et Approches Numériques, LOCEAN/IPSL, F-75005 Paris, France

[2] Department of Marine Sciences, University of Gothenburg, Sweden

[3] DT-INSU Meudon, France

[4] University of Galway, Ireland

*Correspondence to*: kirtana.naeck@locean.ipsl.fr and jacqueline.boutin@locean.ipsl.fr

**Abstract**

The physical and biogeochemical processes governing the air-sea CO₂ flux in the Southern Ocean are still widely debated. The "Southern Ocean Carbon and Heat Impact on Climate" cruise in summer 2022 aimed at studying these processes in the Weddell Sea and in its vicinity. A "CARbon Interface OCean Atmosphere" (CARIOCA) drifting buoy was deployed in January 2022 in the subpolar Southern Ocean, providing hourly surface ocean observations of fCO₂ (fugacity of CO₂), dissolved oxygen, salinity, temperature and chlorophyll-a fluorescence for 17 months. An underwater glider was piloted with the buoy for the first 6 weeks of the deployment to provide vertical ocean profiles of hydrography and biogeochemistry. These datasets reveal an anomalously strong ocean carbon sink for over 2 months occuring in the region of Bouvet Island and associated with large plumes of chlorophyll-a (Chl-a). Based on Lagrangian backward trajectories reconstructed using various surface currents fields, we identified that the water mass reaching the Bouvet Island region originated from the south-west, from the vicinity of sea ice edge in spring 2021. We suggest that a strong phytoplankton bloom developed there in November 2021 through dissolved iron supplied by early sea ice melt in 2021 in the Weddell Sea. These waters, depleted in carbon, then travelled to the position of the CARIOCA buoy. The very low values of ocean fCO₂, measured by the buoy (down to 310 µatm), are consistent with net community production previously observed during blooms occurring near the sea ice edge, partly compensated by air-sea CO₂ flux along the water mass trajectory. Early sea ice retreat might therefore have caused a large CO₂ sink farther north than usual in summer 2022, in the Atlantic sector of the subpolar Southern Ocean. Such events might become more frequent in the future as a result of climate change.



## 1. Introduction

According to the latest Global Carbon Budget, atmospheric $CO_2$ concentrations keep increasing and are projected to be 51% higher than pre-industrial levels in 2023. The ocean helps mitigate the atmospheric $CO_2$ increase by acting as a carbon sink. Indeed, during the past decade (2013-2022), the global ocean absorbed 2.9±0.4 GtC yr$^{-1}$, representing 26% of worldwide $CO_2$ emissions annually (Friedlingstein et al., 2023). The Southern Ocean plays an important role in the ocean's buffering capacity. Defined as the ocean surrounding Antarctica, south of 30-35° S, it covers only about 20-30% of the global ocean surface, yet serves as a main pathway for anthropogenic carbon uptake (Frölicher et al., 2015; Gruber et al., 2019). It is, in fact, responsible for about 40% of the global oceanic uptake of anthropogenic $CO_2$ (DeVries, 2014; Frölicher et al., 2015; Gruber et al., 2019; Mayot et al., 2023).

The direction of the air-sea $CO_2$ flux is determined by the difference between the fugacity of $CO_2$ at the ocean surface, $fCO_2$, and in the atmosphere, $fCO_{2atm}$. When the $fCO_2$ is undersaturated with respect to the atmosphere, the ocean is a carbon sink (Wanninkhof, 2014). Any process that affects $fCO_2$ at local or regional scale, like the biological activity, or the ocean circulation, can therefore affect the ocean carbon pump (Henley et al., 2020). Ultimately, it is the balance between the biological pump, the solubility pump, and ocean circulation which will determine whether the surface ocean will behave as a sink or a source of carbon to the atmosphere.

The lack of $fCO_2$ observations have posed a daunting challenge to the ocean community to understand the full suite of mechanisms controlling the air-sea $CO_2$ flux, and to assess its net magnitude. This is particularly true in the Southern Ocean, which is renowned to be pivotal in controlling global air-sea $CO_2$ flux, but is also one of the regions suffering the most from $fCO_2$ observation scarcity. As a result, there are large discrepancies in the Southern Ocean carbon sink estimates from the literature, which result in large uncertainties in our global estimates of air-sea $CO_2$ flux (Friedlingstein et al., 2023). The difficulty to assess $CO_2$ uptake by the Southern Ocean is exacerbated by the fact that this basin has a large interdecadal variability, and that observation-based products, such as those derived from SOCAT (Bakker et al., 2016), and Global Ocean Biogeochemical Models (GOBMs) have shown different interannual trends for the past years. In this regard, a better understanding of the different processes governing the air-sea flux of $CO_2$ is essential in order to anticipate the effects of climate change on the Southern Ocean's capacity to continue sequestering carbon (Hauck et al., 2023; Mayot et al., 2023; Meijers et al., 2023).

The SO-CHIC ("Southern Ocean Carbon and Heat Impact on Climate") European programme was launched to tackle this knowledge gap (The SO-CHIC consortium et al., 2023). In this context, during the *S.A. Agulhas II* summer research cruise in January 2022 (Ward et al., 2022), a CARIOCA ("CARbon Interface Ocean Atmosphere") buoy was deployed near the Southern Boundary, north-east of the Weddell Sea, and west of Bouvet Island (Naëck et al., 2024). At the same time, an underwater ocean glider (675) (Swart et al., 2024), alongside deep CTD sections (Steiger et al., 2022) were deployed. The glider followed the buoy for six weeks to provide observations of the underlying ocean conditions. The CARIOCA buoy acquired $fCO_2$ measurements in the subpolar region (Fig 1) where surface $fCO_2$ observations have been sparse in the past twenty years (Gruber et al., 2019).

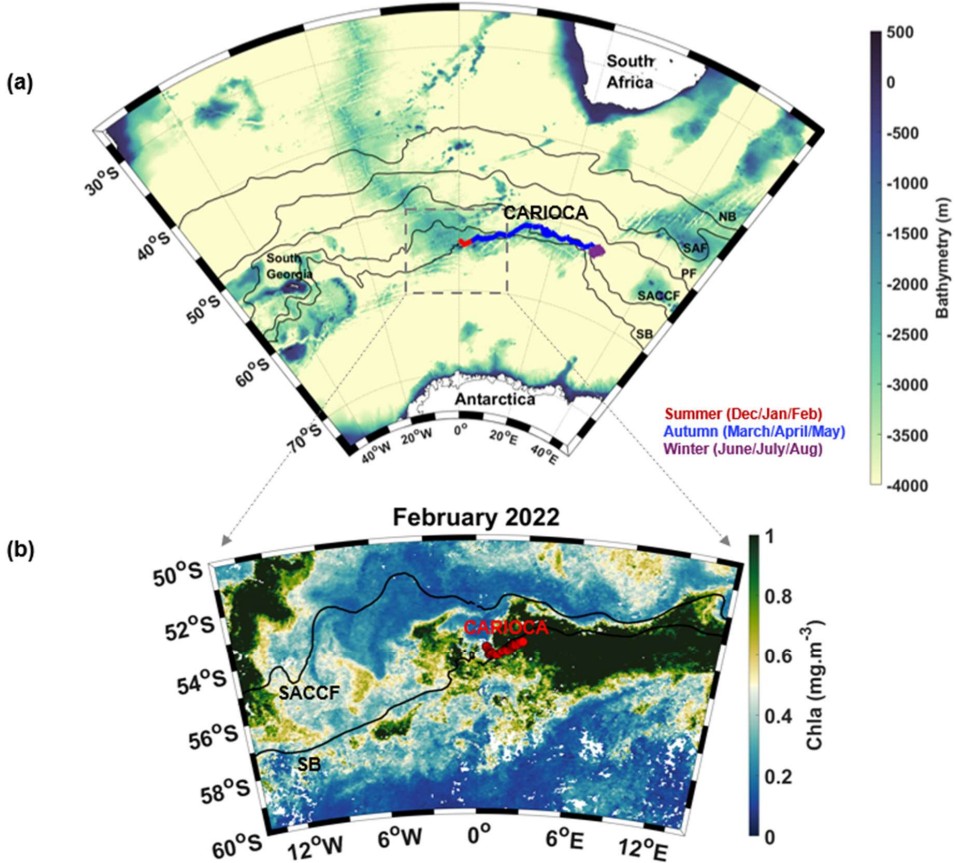


**Figure 1: (a) CARIOCA trajectory from 26 January 2022 to 27 June 2022 superimposed on a bathymetry map of the**
**study zone. Fronts from Park & al. 2019 are indicated, from north to south: Northern Boundary (NB) of the ACC,**
**Subantarctic Front (SAF), Polar Front (PF), Southern Antarctic Circumpolar Current Front (SACCF) and Southern**
**Boundary (SB) of the ACC. (b) CCI Chl-a map with CARIOCA trajectory in February.**
In this paper, we use these unique field observations to investigate the cause of $fCO_2$ variations along the trajectory
of the buoy, with a particular focus on the role of biological activity in shaping these variations. The Southern
Ocean is a High Nutrient Low Chlorophyll (HNLC) region, with iron as the key limiting nutrient for phytoplankton
growth. Sources of iron to the ocean mixed layer therefore condition the efficiency of the biological carbon pump.
Iron can come from island/plateau sources, ocean ridge sources or from an sea ice source (Ardyna et al., 2019). In
summer 2022, west and north-east of Bouvet Island (54.42 °S 3.34 °E, Fig. 1), an unusually large $CO_2$ sink was
observed along the path of the buoy, as well as high biological activity. This study aims to investigate the processes
that contributed to this summer 2022 large $CO_2$ sink. In the next section the instruments deployed, the different
data sets and the methodology used will be described. There will then be a description of the results followed by
a discussion.



## 2. Materials and methods

### 2.1 CARIOCA measurements

The CARIOCA buoy was deployed on 24 January 2022 at 54°S, 0°W. It was anchored at 15 m depth and followed the currents in a quasi-Lagrangian way. A three-wavelength spectrophotometer (434, 596 and 830 nm) was used to measure $fCO_2$. The $fCO_2$ sensor included an exchanger where a dye solution (thymol blue) was brought into equilibrium with seawater via a semi-permeable $CO_2$ membrane. The absorption coefficient of the dye was measured by the spectrophotometer and was then related to the carbonate properties of seawater. The three-wavelength measurements enable correction of any modification of the optical path or of the opacity of the optical cell (Copin-Montégut et al., 2004). The absolute precision of the $fCO_2$ is estimated ± 3 µatm and its relative precision is ± 1 µatm. A thermosalinograph measured the sea surface temperature at 2 m depth, SST, and the conductivity, from which sea surface salinity, SSS, was derived. An optode measured dissolved oxygen ($O_2$). According to the manufacturer, the precision of the $O_2$ measurements is ~ 8 µmol $L^{-1}$ (5 %). Calibrated values were given by the manufacturer for pure water and they were corrected from the effects of temperature and salinity as described in Merlivat et al. (2015). A fluorometer measured the fluorescence which was calibrated in Chl-a units using the calibrated Seaglider (675) fluorescence (see section 2.3). An anemometer measured wind speed and a barometer measured atmospheric pressure (Patm) at 2 m above the sea surface. The wind at 10 m above the ocean was derived assuming a neutral atmosphere. As from 4 June 2022, the CARIOCA atmospheric sensor stopped working, and the wind and atmospheric pressure data were replaced with data from ERA5 (C3S, 2018). During their common period, CARIOCA and colocated ERA5 data were very similar (mean difference = 0.26 m $s^{-1}$, $R^2$ = 0.86, not shown). The buoy acquired 17 months of data which was transmitted in real time via ARGOS, and stopped functioning on the 24 June 2023. In this study, we only use observations from the period of 26 January 2022 to 27 June 2022, which corresponds to the time during which a strong $CO_2$ sink was observed.

### 2.2 CARIOCA derived parameters

The air-sea flux of $CO_2$ represents the exchanges of $CO_2$ at the ocean-atmosphere interface. This air-sea flux of $CO_2$ (F) was calculated using Wanninkhof's methodology (Wanninkhof, 1992, 2014).

$$F = K (fCO_{2ssw} - fCO_{2atm}) \tag{1}$$

where $fCO_{2ssw}$ and $fCO_{2atm}$ are the fugacity of $CO_2$ at the ocean surface and in the atmosphere and K is the $CO_2$ exchange coefficient (Wanninkhof, 1992). The $fCO_{2atm}$ was determined using atmospheric $CO_2$ concentration data ($xCO_2$ atm) from SPO ("South Pole Antarctica"), and CARIOCA Patm according to Eqn. 2 in Boutin et al. (2008). To eliminate $fCO_2$ changes linked to temperature and salinity, the concentration of dissolved inorganic carbon (DIC) was estimated. At given SST and SSS, the DIC and alkalinity was estimated from $fCO_2$. The alkalinity was calculated according to Lee et al. (2006). The Matlab routine *CO2SYS*, with dissociation constants K1 and K2 of Lueker et al. (2000) was used to derive DIC. Oxygen saturation ($O_{2sat}$), the degree of oxygen saturation ($pO_{2sat}$) and the oxygen flux at the air-sea interface ($FO_2$) were calculated as per Merlivat et al. (2015). The oxygen anomaly was calculated by subtracting the dissolved oxygen concentration calculated at saturation from the dissolved oxygen concentration ($O_2 - O_{2sat}$). During photosynthesis, a positive anomaly is expected whereas during respiration / remineralization, a negative anomaly is expected. The oxygen measurements of the CARIOCA drifter could not be recalibrated at sea, because unfortunately the $O_2$ measured by the CTD at the deployment was not calibrated.



An empirical correction of +8 µmol/kg (which corresponds to the precision of the optode) was applied to bring the values of $O_2$–$O_{2sat}$ above zero, during periods of high biological activity. These high biological periods are detected from CARIOCA fluorescence and opposite variations in $O_2$ and DIC. The carbon net community production, NCPc, and the oxygen net community production, NCPo$_2$, were estimated during periods of high biological activity, during which diurnal cycles with DIC and $O_2$-$O_{2sat}$ in opposition were observed by the CARIOCA buoy, following the same methodology as in Merlivat et al. (2015), using mixed layer depth (MLD) derived from glider vertical profiles (see section 2.3).

**2.3 Sealider measurements and derived parameters**

The Seaglider (SG675) followed the CARIOCA buoy for a month and a half, from 31 January 2022 to 10 March 2022 (39 days) (Fig. 1), providing vertical profiles of temperature, salinity, oxygen and fluorescence of the upper 1000 m of the water column. There was a time lapse of about one day between CARIOCA and the glider, the glider being about one day behind the buoy each time. Both instruments were less than 20 km apart during that period. For this study, only profiles between the 31 January and the 10 March 2022 were considered, which was when the glider and the CARIOCA were nearest to each other. The glider data was processed using the "GliderTools" package (Gregor et al., 2019; Swart et al., 2024). For the salinity data, outliers and spikes were removed and a smoothing was applied with Savitzky-Golay filter. The glider salinity data was also validated using the CTD salinity profile at the time of deployment (on the 23 January 2022 14h46). The CTD salinity data had already been calibrated, and the CARIOCA SSS data was in turn validated using the already validated glider surface salinity data. The glider salinity at 2 m and the CARIOCA SSS (at 2 m), were in good agreement, with a small difference of about 0.05 pss. Using "GliderTools", the glider fluorescence profiles were also quality controlled. Outliers, spikes and bad profiles were removed, and the data was corrected for in situ dark counts. A quenching correction using the same method as Thomalla et al., 2018 was applied. The glider fluorescence was calibrated and converted to chlorophyll-a using the CTD fluorescence at the time of deployment.

The mixed layer depth, MLD, was estimated using the glider data using the density threshold of 0.03 kg m$^{-3}$ (de Boyer Montégut et al., 2004)). It has been observed in past studies that DIC can be more sensitive to the mixing layer, XLD, which can be determined using turbulence measurements (Nicholson et al., 2022; Pellichero et al., 2020; Sutherland et al. 2014; Giunta and Ward 2022). The empiric relationship approximated by Nicholson et al. (2022) was used to estimate XLD. (The MLD and the XLD are surface layers directly influenced by the atmosphere. The MLD has homogenised, formed as a result of mixing, while the XLD is still actively influenced by turbulence and still mixing).

**2.4 *S.A Agulhas II* TSG**

The S.A. *Agulhas II* was equipped with a thermosalinograph (TSG) (Ward et al., 2024), and ship measurements of salinity and temperature were collected, from South Africa to Antarctica, and back again, from the 3 December 2021 to the 28 January 2022. The *S.A. Agulhas II* TSG SSS data was calibrated using underway water samples collected during the SO-CHIC cruise (Ward et al., 2022).



**2.5 Satellite, analysis, and reanalysis datasets**

Chlorophyll-a satellite images are from the Climate Change Initiative product version 6.0 (OceanColour - CCI, 2024), from 1997 to 2022, for spring (November: 1997-2020) and summer (January, February: 1998-2022). Monthly and weekly means were used for the Chl-a. For the surface salinity data, Glorys reanalysis (Global Ocean Physics Reanalysis, 2024), Mercator analysis (Global Ocean Physics Analysis and Forecast, 2024) at 5 m depth, SMOS CATDS CECv9.0 (18 days) satellite data (Boutin et al., 2023) and ISAS analysis at 5 m depth (In Situ Analysis System, an optimal interpolation from ARGO floats) (Szekely et al., 2024) were used. A colocalization between Mercator / Glorys salinity data and CARIOCA SSS indicates that both datasets were in good agreement, showing similar patterns, but with Mercator / Glorys salinity data being lower than the CARIOCA by about 0.1 pss from February 2022 to mid-April 2022 (Fig. B1 in Appendix).

Daily and monthly data, from Mercator analysis were used as of July 2021, and Glorys reanalysis was used for the years before July 2021. The Mercator/Glorys products were also used to analyse vertical salinity and surface current velocities. Sea ice data was obtained from OSI SAF (EUMETSAT - Product Navigator - Global Sea Ice Concentration Climate Data Record v3.0 - Multimission, 2024; EUMETSAT - Product Navigator - Global Sea Ice Concentration Interim Climate Data Record Release 3 - DMSP, 2024).

**2.6 Lagrangian analysis**

Backward trajectories of water masses were computed with a Lagrangian analysis (Sergi et al., 2020) using the LAMTA software (Rousselet, L et al., 2024 pre-print). After initialising the particles at specific dates, longitudes and latitudes, a backward advection was performed to estimate the trajectories of these particles several months before. To determine which water masses reached the CARIOCA, particles were initialised at the CARIOCA coordinates, with one coordinate point per hour (i.e. 24 points advected backwards in time each day). Three current products were tested, namely, GlobCurrent (Global Total (COPERNICUS-GLOBCURRENT), Ekman and Geostrophic currents at the Surface, 2024), Mercator Analysis (Global Ocean Physics Analysis and Forecast, 2024) / Glorys reanalysis (Global Ocean Physics Reanalysis, 2024) and OSCAR (OSCAR third degree resolution ocean surface currents | PO.DAAC / JPL / NASA, 2024). Using Mercator analysis / Glorys reanalysis, different depths were also tested, namely 5 m, 15 m, 34 m and 55 m. Using OSCAR, the backward trajectories in November, January and February were also computed, from 1997 to 2022. In the next sections, we are showing the results obtained using the total currents, i.e. the sum of the geostrophic and Ekman currents, at 15 m depth. The reliability of numerical trajectories for reconstructing phytoplanktonic blooms from nutrient sources and up to mesoscale precision has been validated in the Southern Ocean from multi satellite data, surface drifters, and lithogenic isotopes (Sergi et al. 2020; d'Ovidio et al. 2015; Sanial et al. 2014).





## 3. Results

### 3.1 CARIOCA and Seaglider observations

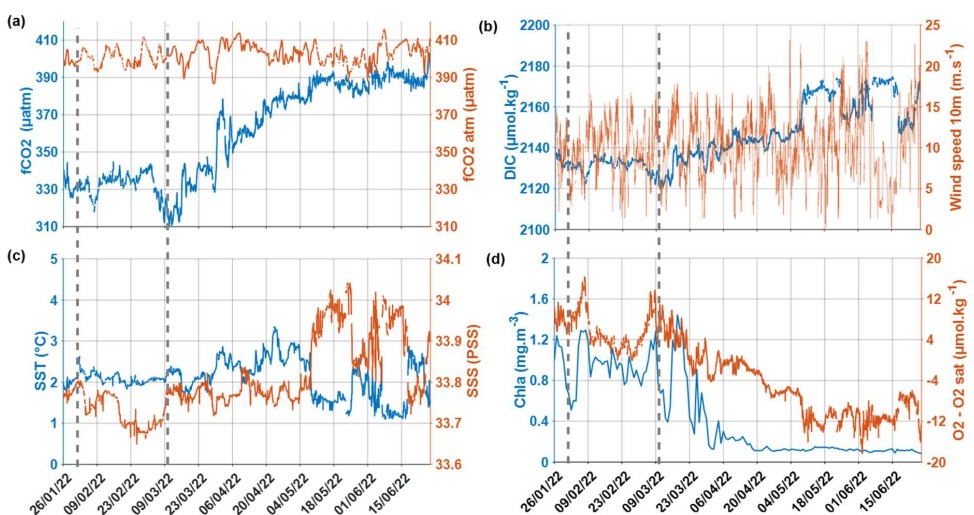

**Figure 2: CARIOCA time series from the 26 January to the 27 June 2022: (a) Atmospheric and surface ocean fCO₂**
**(fCO$_{2atm}$ and fCO₂), (b) DIC and wind speed, (c) SST and SSS, (d) Chl-a and O$_2$-O$_{2sat}$, with the period during which the**
**glider followed the buoy (31 January to 10 March 2022) indicated by dotted lines.**

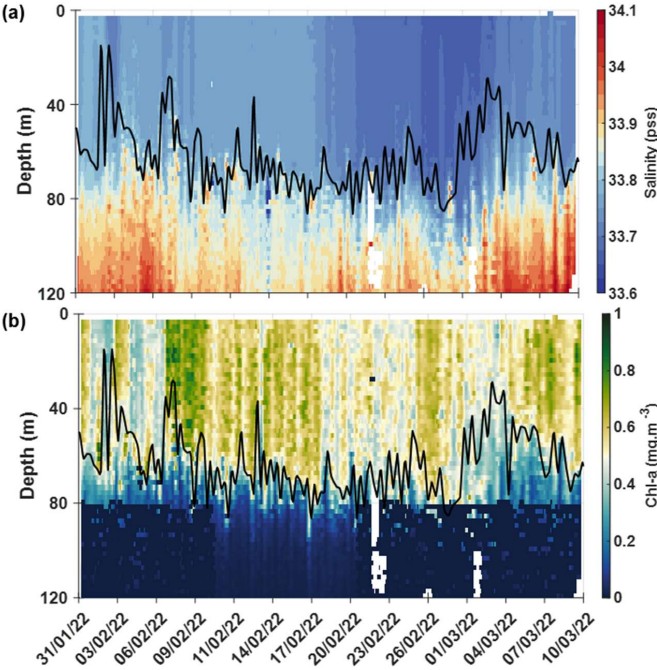

**Figure 3: Seaglider profile time series between the 31 January and the 10 March 2022 of (a) salinity, (b) Chl-a, with**
**the MLD indicated with the black line.**





Low surface salinities were measured in February 2022, both by the CARIOCA buoy (Fig. 2 (c)) and by the glider
(Fig. 3 (a)) with minimum values of salinity in the second half of February. The CARIOCA measured surface
salinities around 33.7 pss between 17 February and 8 March 2022. Both the glider and the CARIOCA observed
maximas of surface Chl-a around the 7 February and the 7 March 2022 (Fig. 2 (d) and Fig. 3 (b)). According to
the glider profiles, the low values of salinity, and high concentrations of Chl-a, observed from early February to
early March, were well mixed within the mixed-layer, confined to the top 60 m or so (Fig. 3). According to fig. 2
(a), a very low $fCO_2$ was observed by the CARIOCA in summer 2022, while the buoy was in this fresh water mass,
enriched in Chl-a and oxygen. The $dfCO_2$ is about -60 µatm from the end of January to the end of March 2022.
Minima of $fCO_2$ (~ 320 µatm) and of DIC (~ 2120 µmol kg$^{-1}$) occurred on the 7 February and the 7 March 2022,
which also coincided with maximum concentrations of chlorophyll-a (~ 1.3 mg m$^{-3}$) and of oxygen. Around these
dates, diurnal cycles of DIC and $O_2$-$O_{2sat}$, with the DIC and $O_2$-$O_{2sat}$ in opposite phases were clearly observed, and
allowed NCP estimates (see next section). According to CARIOCA measurements (Fig. 2 (a)), after March, as the
CARIOCA drifted away from the phytoplankton bloom, $fCO_2$ started increasing, until it reached values close to
equilibrium with the atmosphere on 27 June 2022.
**3.2 NCP and impact of wind on MLD**

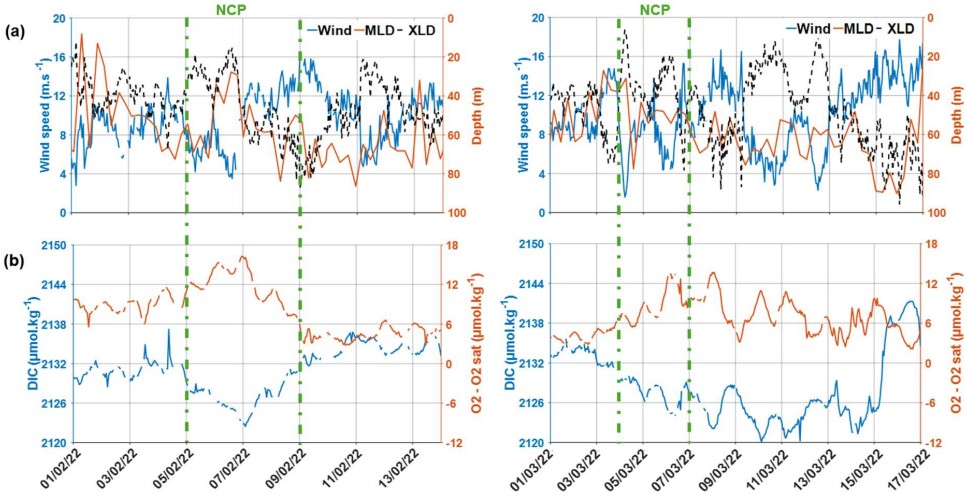


**Figure 4: CARIOCA time series, during NCP periods on the 5-9 February 2022 (left), and on the 4-7 March 2022**
**(right), (a) Wind speed, MLD and XLD (b) DIC and $O_2$-$O_2$ sat.**
During the two periods dominated by biological activity, on the 5-9 February 2022, and on the 4-7 March 2022,
Chl-a increased by 0.5 mg m$^{-3}$ and DIC decreased by 10 µmol kg$^{-1}$ (Fig.2, Fig. 4 and Fig. A1 in Appendix A). Just
before the peaks of Chl-a (corresponding to minimum concentrations of DIC and maximum concentrations of
oxygen), there were very low wind speeds, associated with a shoaling of the mixed layer (Fig. 2 and Fig. 4), which
itself might have enhanced Chl-a bloom at the surface (Fig. 3). During the February event, NCPc was estimated
to reach -91 mmol m$^{-2}$ d$^{-1}$, while NCPo$_2$ was 132 mmol m$^{-2}$ d$^{-1}$. During the March event, NCPc of -104 mmol m$^{-}$
$^2$ d$^{-1}$ and NCPo$_2$ of 138 mmol m$^{-2}$ d$^{-1}$ were estimated (Fig. A1 in Appendix). It is interesting to note that the two



values of the photosynthetic quotient, PQ= NCPo$_2$/ NCPc, respectively equal to 1.45 and 1.33 are in close good agreement with the value 1.4 expected in a new production regime (Laws, 1991).

After the 7 February and the 15 March 2022, there was a strengthening of wind speeds, the XLD became deeper than the MLD, probably entraining waters rich in DIC from the subsurface layer. Indeed, the CARIOCA measured higher concentrations of DIC at the same time as higher wind speeds, and lower concentrations of O$_2$-O$_{2sat}$ and of Chl-a (Fig. 2 and Fig. 4). This might suggest that mixing events driven by high winds sometimes compensated for the CO$_2$ undersaturation driven by biological activity. However, the mixing events shown here are only on synoptic time scales. For instance, after the 10 February 2022, the wind decreased again and DIC concentrations stopped increasing (Fig. 4 (a)). Overall, the CARIOCA measured low fCO$_2$ and DIC concentrations coinciding with high concentrations of Chl-a and oxygen, during the whole summer 2022. This suggests that biological activity was the dominant driver of the DIC seasonal variation, but the overall DIC concentration might have been even lower without wind driven mixing.

**3.3 Origin of fresh water mass and phytoplankton bloom**

In summer 2022, the CARIOCA and glider were found in productive waters of low salinity (Fig. 2 and 3). The presence of this low salinity upper ocean was also confirmed by the *S.A Agulhas II* thermosalinograph SSS observations. Moreover, SSS images from the SMOS satellite mission, as well as salinity estimates, at 5 m, from an ocean analysis (Mercator) and reanalysis (Glorys), and from an observational based gridded ocean salinity estimate (ISAS), are all in agreement with the in-situ measurements and show low salinities in that region, in summer 2022 (Fig. B2 in Appendix B). This surface fresh layer suggests a stratification of water masses, which might have favoured the sustained growth of phytoplankton in a shallow mixed layer, where more light was available.

To investigate the origin of these waters, we computed backward trajectories of virtual particles that were released at the CARIOCA location. These particles were advected backward in a suite of different velocity field estimates, at 15 m depth (see Methods). According to salinity maps from Mercator/Glorys, and to backward trajectories spanning several months before the 2022 deployment of the instruments, the fresh water mass in which the CARIOCA drifted was advected from a region south-west of their position, in the Weddell Sea. The formation of this fresh water mass can be seen, from September to December 2021, using salinity maps and sea ice data from OSI SAF (Fig. C1 in Appendix). The decrease in salinity started near the South Sandwich trench, around 25° W and 60° S, near the sea ice edge in September 2021, when the sea ice started to retreat. It continued to develop eastward near the sea ice edge until November 2021 (Fig. C1 in Appendix). Fig. 5 (b) shows the sea ice retreat that month and the decrease in salinity that is very likely due to ice melt. The water mass then travelled north-east (Fig. C1 and Fig. E1 in Appendix). In March 2022, the waters reaching CARIOCA still originated from the south-west but then passed near Bouvet Island, before reaching the CARIOCA (Fig. E1 in Appendix). This corresponds to a decrease in fCO$_2$, DIC and increase in Chl-a (Fig. 2).



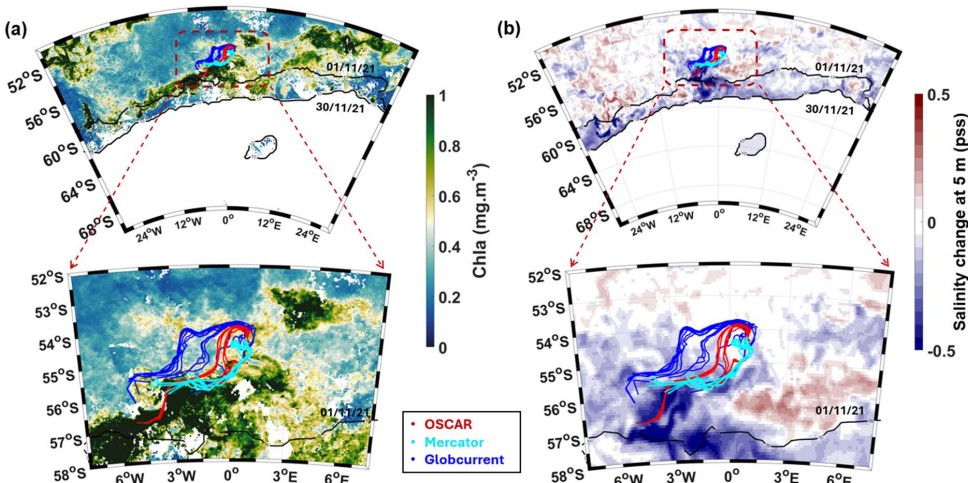

**Figure 5: (a) CCI Chl-a in November 2021, with sea ice mask (b) Mercator salinity change at 5 m in November 2021 (difference between the 30 November and 1 November), with black lines representing the sea ice concentration at 10%, on the 1 November 2021 and on the 30 November 2021. Bottom figures represent a zoom of the region, with backward trajectories (using Mercator, OSCAR and GlobCurrent total currents at 15 m depth) from the 7 February 2022 (from CARIOCA's location) to the 1 November 2021.**

On Fig. 5, backward trajectories, computed using currents from OSCAR, Mercator / Glorys and GlobCurrent, show that waters found at CARIOCA location (0.8° E, 54° S) on the 7 February 2022, came from a large phytoplankton bloom (high Chl-a) that occurred in a region where surface salinity decreased at around 4-5 °W and 55.5-56.5 °S, near the sea ice edge, on the 1 November 2021. This corresponds to an export of waters from the Weddell Sea, south of 55° S (Vernet et al., 2019). No satellite Chl-a data is available in October 2021 due to clouds; however we can suppose that there was less sunlight available in October and that the phytoplankton bloom formed in November 2021.

### 3.4 Comparison between spring-summer 2021-2022 and 2018-2019

Previous campaigns were conducted at the same season, near 54°S-0°W, in 2019 (Nicholson et al., 2022; Ogundare et al., 2021). A comparison was therefore made between the summers 2022 and 2019. On 19 December 2018, a Wave Glider (a surface autonomous vehicle) was deployed, at 0° E, 54° S (Nicholson et al., 2022). It stayed fixed at this position for about two months, and performed surface measurements, until 11 February 2019. A comparison was made between this Wave Glider and the CARIOCA surface data. The CARIOCA stayed near 0° E, 54° S, from 26 January to 1 February 2022.

In 2019, the $fCO_2$ measured by the Wave Glider was near equilibrium with the atmosphere (2019 Wave Glider $fCO_2 \sim 410$ µatm) whereas the CARIOCA buoy measured a large $CO_2$ undersaturation (2022 CARIOCA $fCO_2 \sim 330$ µatm, Fig. 2a). In summer 2022 (from 26 January to 11 February 2022) the MLD was around 60 m, whereas in summer 2019 (from 19 December 2018 to 11 February 2019) the MLD was deeper, around 100 m (not shown). According to Nicholson's study, in 2019 the DIC concentration was around 2180 µmol kg$^{-1}$, while in January to



March 2022 it was much lower (~ 2130 µmol kg$^{-1}$). After March, as the buoy went out of the Chl-a rich waters,
the CARIOCA DIC increased gradually, until it reached concentrations around 2170 µmol kg$^{-1}$ in June 2022 (Fig.
2 (b)), close to the ones measured by the Wave Glider in 2019.

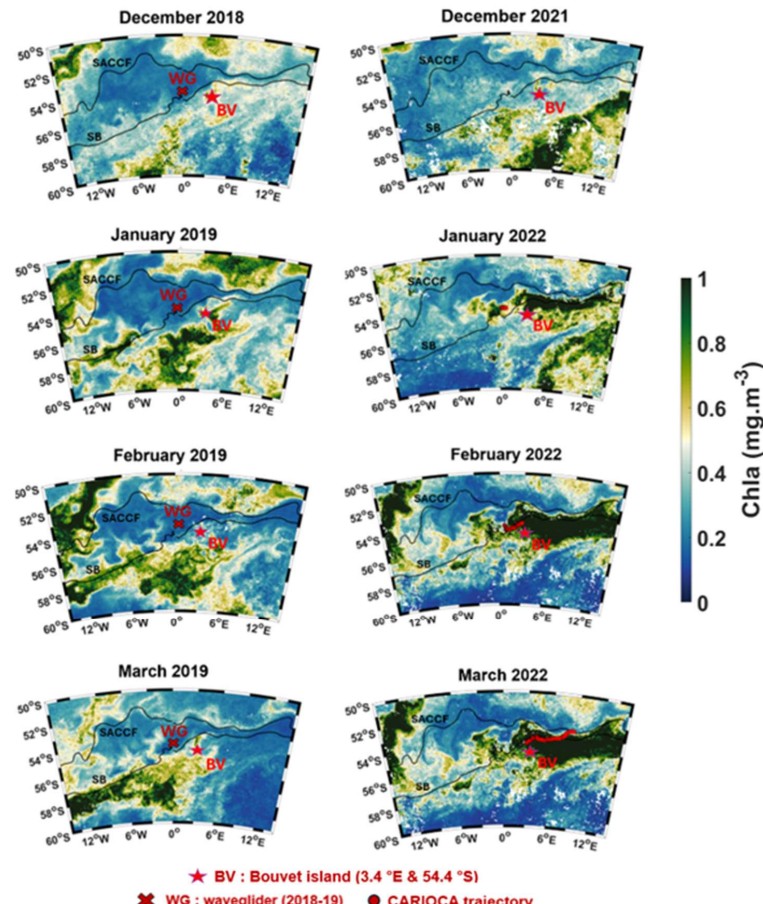


**Figure 6: CCI Chl-a maps from December to March, left panel: 2018-2019, with waveglider position for the given month**
**in red, right panel: same as left, but for 2021-2022, with CARIOCA position for the given month in red. (SB: Southern**
**Boundary, SACCF: Southern Antarctic Circumpolar Current Front).**

For the Chl-a comparison, CCI data were used to determine the Chl-a concentrations in the region in summer 2019
and 2022 (Fig. 6). From January to March 2022, a large phytoplankton bloom was observed around the Southern
Boundary, near the CARIOCA position (Fig. 6 (b)). This phytoplankton bloom was not present at the Wave Glider
and CARIOCA locations, in summer 2019 (Fig. 6 (a)). Indeed, around 0° E, 54° S, in 2019, there was no
chlorophyll-a. So, for different years, at the same position and for the same period, both instruments sampled
similar water masses but there was a large $CO_2$ undersaturation in summer 2022 not present in 2019. This is then
likely to be related to the large phytoplankton bloom present, at CARIOCA location, in January 2022. The
phytoplankton was farther south in January-March 2019. Indeed, in 2019, the phytoplankton stayed south of the
Southern Boundary (SB). The Chl-a concentrations were also lower in summer 2019 compared to 2022, and





reached up to 1.5 mg m$^{-3}$ in 2022, whereas in 2019, the maximum Chl-a concentrations did not exceed 1 mg m$^{-3}$.
In contrast, in January 2022, the CARIOCA entered in a phytoplankton bloom at 0° E, 54° S. The buoy then
continued to sample waters rich in Chl-a in February and March 2022.

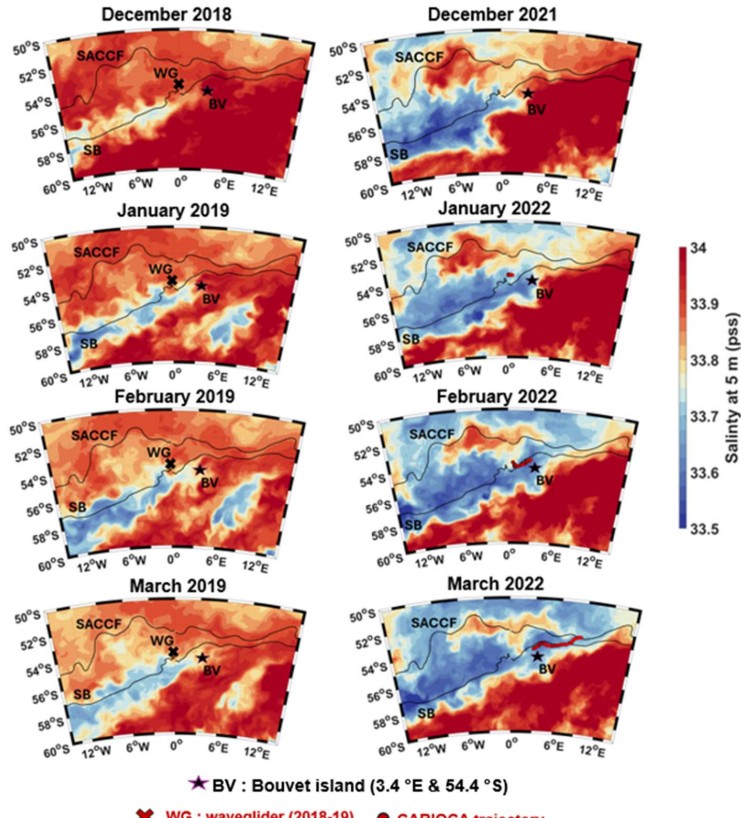


**Figure 7: Mercator / Glorys salinity at 5 m, from December to March, left panel: 2018-2019, with waveglider position**

**for the given month in red, right panel: 2021-2022, with CARIOCA position for the given month in red. (SB: Southern**

**Boundary, SACCF: Subantarctic Circumpolar Current Front).**

The phytoplankton bloom in summer 2022 in fact coincides with the presence of a fresh water mass. Mercator
salinity profiles for spring 2021-summer 2022, show that this lower salinity was only present in the first 60 m,
which also corresponds to the glider salinity profile. According to Mercator, in 2019, the lower salinities were
much farther south (Fig. 7).
A comparison was also performed with surface transects of the Norwegian RV *Kronpins Haakon* cruise from the
28 February 2019 to the 10 April 2019, (Ogundare et al., 2021). A ship transect crossed the CARIOCA path at
around the same dates (end of March/beginning of April for both), for different years (2019 and 2022). North of
the Southern Antarctic Circumpolar Current Front (SACCF), in April, the DIC, SST and SSS values for both years
were consistent with each other. South of the SACCF, in 2019 the DIC concentration was higher (~ 2150 µmol
kg$^{-1}$) compared to the CARIOCA DIC concentration in 2022 (as low as 2120 µmol kg$^{-1}$, South of the SACCF).
The ship's path in 2019, south of the SACCF, was between 5-8° E and 53-57° S, where the Chl-a concentrations



were low (Fig. 6). At these positions, higher salinities and temperatures were also measured by the RV *Kronpins*
*Haakon*, in 2019 (SSS ~ 34 pss before April 2019) (Fig. D1 in Appendix).
**4. Discussion**
In summer 2022, the CARIOCA buoy crossed a massive phytoplankton bloom (Figure 1) and measured an
unusually large $CO_2$ sink near the Southern Boundary and Bouvet Island, east of 0° E, 54° S (Figure 2a). According
to pCO2 climatologies (Takahashi et al., 2012) or recent in situ measurements in 2019 (Nicholson et al., 2022;
Ogundare et al., 2021), this region is usually in equilibrium with the atmosphere or is a much smaller carbon sink
than the one observed in summer 2022. Sergi et al. (2020) showed that moderate bloom can develop around Bouvet
Island by a combination of fertilization from the island itself and from the proximity of seamounts acting as
hydrothermal sources of iron. However, the bloom observed in summer 2022 (Fig. 1b) was more massive. In fact,
it was the largest bloom observed over the past twelve years (Fig. 8a) (and also over the last 25 years, not shown),
as large as those typically observed on the Kerguelen Plateau (Blain et al., 2007).

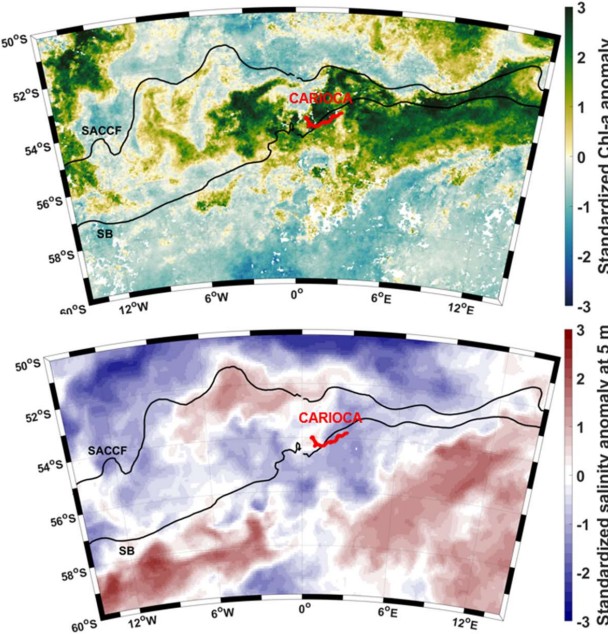


**Figure 8: Anomalies in February 2022, relative to a 2011-2022 climatology, normalised with the standard deviation of**
**monthly values for: (a) Chl-a and (b) Salinity at 5 m using Mercator analysis. CARIOCA trajectory in February (red).**
Moreover, the Lagrangian backward trajectories for the January-February 2022 bloom are not consistent with an
origin from Bouvet Island or from seamounts, in contrast with the case in Sergi et al. (2020). Another possible
source of iron fertilization could come from the melting of icebergs (Person et al., 2019). An analysis of satellite
images enables us to rule out the hypothesis of iceberg melting in the vicinity of the CARIOCA drifter, or of water
fertilised by the melting of the supergiant iceberg A68A (Smith and Bigg, 2023) that happened near South Georgia
in Fall 2021, being the cause of the bloom. Although there were some iceberg fragments present near the
CARIOCA after its deployment, in January 2022 (Fig. S2 in Supplement), the large size of the fresh water mass



335 coming from the south-east, all point towards an export of fresh waters from the Weddell Sea. Salinity was

336 anomalously fresh (Fig 8b), although there was no significant rainfall during that period at the position of the fresh

337 water mass (not shown). Both salinity maps (from Mercator/Glorys) and backward trajectories enable us to track

338 the travel of this fresh water mass, from its formation in spring 2021, to the position of the CARIOCA and glider,

339 in summer 2022. The fresh water mass was formed near the sea ice edge (from September 2021 to November

340 2021, Fig. C1). Indeed, in 2021, the ice retreated abnormally early, from September to mid-November 2021,

341 anomalous north-eastward motion of sea ice was observed in the north-east Weddell Sea (Wang et al., 2022, their

342 Fig. 3 and 4) and the austral summer 2022 marked a record for the lowest sea ice extent in Antarctica (Wang et

343 al., 2022).

### 4.1 Impact of anomalous sea ice retreat on phytoplankton blooms

345 Previous studies have shown that the melting of sea ice can fertilise the region near the sea ice edge and stratify

346 the water masses (Briggs et al., 2018; McClish and Bushinsky, 2023). We can suppose that this is what promoted

347 the development of a phytoplankton bloom at the sea ice edge in spring 2021 (Fig. 5a). Model results of Death et

348 al. (2014) also suggested that subglacial meltwaters may constitute a significant additional source of bioavailable

349 iron to the Southern Ocean, supplementing iceberg sources, and that the impact of the glacial iron flux may be

350 large, extending across much of the Southern Ocean due to the redistribution of the iron by ocean circulation.

351 We can thus infer that phytoplankton bloom near CARIOCA trajectory in January-February 2022 was favoured

352 by stratified, fresher waters (Fig. 8b) and micro nutrients supplied by ice melt and redistributed by ocean

353 circulation. This is supported by fresh and enriched Chl-a water that detached from the sea ice edge region around

354 4°W in November 2021 and migrated north-westward (Fig. 5). This leads to the presence, in December 2021, of

355 a patch of moderate Chl-a around 6°W 57°S (Fig. 6) along the Lagrangian backward trajectory (Fig. E1 in

356 Appendix). Then, this patch migrates north-eastward, and reaches the CARIOCA in January 2022, with Chl-a

357 slightly higher than in December, possibly because of additional fertilization by nearby seamounts or because of

358 the presence of different phytoplankton community composition. In March 2022, the waters reaching CARIOCA

359 originated from farther south, closer to the ice (Fig. E1 in Appendix), then passed near Bouvet Island, before

360 reaching the CARIOCA (Fig. E1). Hence these waters might have been even more enriched in micro nutrients

361 related to sea ice melt. Also, as shown by previous studies, the proximity of Bouvet Island might have further

362 fertilised the region (Sergi et al., 2020). This is consistent with the decrease in DIC and increase in Chl-a observed

363 by CARIOCA from the beginning of March (Fig. 2).

364

### 4.2 Interannual variation

366 The backward trajectories show that waters near 0°S, 54°S in February, generally came from the south-west the

367 previous years as well (Fig. S4). We therefore investigate a possible link between the intensity of the bloom at the

368 extremities of the Lagrangian backward trajectory between November and February, from 2010 to 2022, to study

369 a possible link between spring blooms near the sea ice edge and summer blooms near Bouvet Island.

370 In spring 2021-summer 2022, the blooms at both extremities of the backward trajectories were much more

371 pronounced than usual, with Chl-a concentrations above 1 mg.m$^{-3}$ in both November 2021 (along the backward

372 trajectory) and in February 2022 (along the CARIOCA trajectory) (Figure 9).





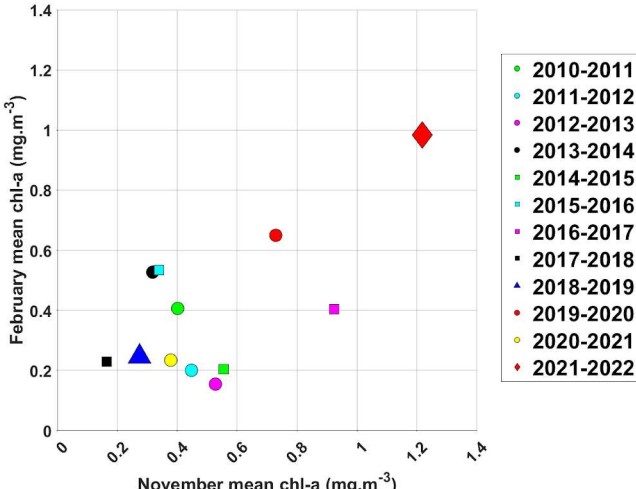

373

**Figure 9: Mean Chl-a along CARIOCA position in February plotted against the mean Chl-a along the backward**

**trajectories (computed using OSCAR currents) in November, for each year from 2010 to 2022.**

This supports the hypothesis that there might be a link between the fertilization near the sea ice edge and the biological activity at the CARIOCA location. This also suggests that a strong spring bloom, promoted by sea ice retreat, could promote the development of a strong summer bloom farther north. Following the same logic, when low concentrations of Chl-a were observed in November, similarly low concentrations of Chl-a were usually observed in February (see for instance 2018-2019, Fig. 9).

**4.3 Interpretation of the low DIC and fCO₂ observed by CARIOCA**

During sea ice retreat, large net community production occurs in the seasonal ice zone (SIZ), likely due to iron delivery or low grazing rates (McClish and Bushinsky, 2023). Moreover, these authors found, based on BGC Argo floats observations, that the bloom NCP increases in case of early sea ice retreat. Given that the water mass reaching the CARIOCA subpolar region remained in the bloom near sea ice edge from September to November 2021, i.e. over the whole bloom period, it seems reasonable to consider that the DIC of this water mass was affected by a bloom NCP typical of early ice retreat, of 3 molC m$^{-2}$ (Figure 4 of McClish and Bushinsky, 2023) which would correspond to a fCO$_2$ as low as 240 µatm (Appendix F). Assuming that from end November 2021 to January 2022, the fCO$_2$ change along the water mass trajectory was only affected by air-sea exchange and temperature change, we estimate a fCO$_2$ of 342 µatm at the end of January 2022 (see details in Appendix F), a value very close to the one observed by CARIOCA (Figure 2a). In this crude estimation, we neglect biological production along the water mass trajectory and mixing with subsurface water. The latter seems at first order reasonable given that the salinity anomaly persisted along the water mass trajectory until January 2022. For what concerns the biological activity, the Chl-a was much lower along the water mass trajectory in December 2021-January 2022 than in the vicinity of the sea ice edge in spring 2021, suggesting low NCP. Moreover, the DIC derived from CARIOCA observations in January-February 2022 remained relatively stable suggesting a balance between biological production and other processes, such as subsurface mixing or consumption by grazers.



**4.4 Study limits and perspectives**

Our Chl-a study was focused only on the region surrounding Bouvet Island, from 50° S to 60° S, and 15° W to 15° E. We could envision further studies on the impact of sea ice retreat at latitudes farther north, in the whole Southern Ocean, not just in the Atlantic sector. The kind of analysis performed in this study could be extended on other islands in the Southern Ocean and see if the combined effect of sea ice retreat and fertilization from the island, could enhance phytoplankton development around other Southern Ocean islands. The year 2022 was a record for lowest sea ice extent in austral summer (Wang et al., 2022). We could hypothesise that the productive waters travelled farther north than usual, due to this unprecedented low sea ice extent record. It has been shown that due to global warming, there is a tendency for more sea ice retreat in the Southern Ocean in the future (Wang et al., 2022). This could lead to stronger undersaturation of $fCO_2$ in regions usually in equilibrium with the atmosphere in austral summer. Phytoplankton travelling to higher "northern" latitudes could also lead to a shift in the hunting zones of predators and might therefore have an impact on the ecosystem, in the long term. Studies have shown that due to higher emissions and a warmer climate, there might be a change in the future location, mechanism and seasonality of the carbon sink in the Southern Ocean, with an increase of $CO_2$ uptake in certain regions (Hauck et al., 2015; Mongwe et al., 2024). In this context, events such as the one observed in summer 2022, that is, an increased sea ice retreat leading to a shift of the $CO_2$ sink to higher "northern" latitudes, should become more frequent, and should be monitored carefully.

**5. Conclusion**

Lagrangian backward trajectories suggest that the summer 2022 phytoplankton bloom in the subpolar ocean is mainly due to early sea ice retreat the previous spring and not due to the proximity of Bouvet Island or seamounts, as was the case in Sergi et al. (2020). Although there was probably some contribution, in March 2022, due to fertilization from Bouvet Island, the CARIOCA buoy was already in Chl-a rich waters, at the end of January 2022, west of Bouvet. Indeed, the phytoplankton bloom in which the CARIOCA entered did not originate from Bouvet Island but coincides with the presence of a fresh water mass. Mercator/Glorys salinity maps and Lagrangian backward trajectories, from November 2021 to March 2022, show that this fresh water mass sampled by the buoy and the glider, originate from the vicinity of sea ice edge, south-west of the instruments' deployment, at around 4-5° W and 55.5-56.5° S. Early sea ice retreat in spring 2021 fuelled seawater with micro nutrients leading to a large phytoplankton bloom near sea ice edge, which then travelled to the CARIOCA position. These waters were likely already depleted in carbon near sea ice edge when they escaped towards CARIOCA location which led to very undersaturated CARIOCA $fCO_2$. This shows that productive waters from the sea ice edge travelled farther north than usual, up to 54° S, causing a strong undersaturation of $fCO_2$ in regions usually known to be in equilibrium with the atmosphere. Austral summer 2022 was a record for an unusually low sea ice extent (Wang et al., 2022). Our observations highlight the northward migration of the $CO_2$ sink associated with early sea ice retreat, a phenomenon projected by Earth System Models under climate change (Mongwe et al. 2024), with significant reduction in biologically-derived $fCO_2$ associated with melting ice, and the consequent increase in the oceanic $CO_2$ sink in certain sectors of the Southern Ocean.



**Appendix**

**Appendix A: Net Community Production (NCP) estimations**

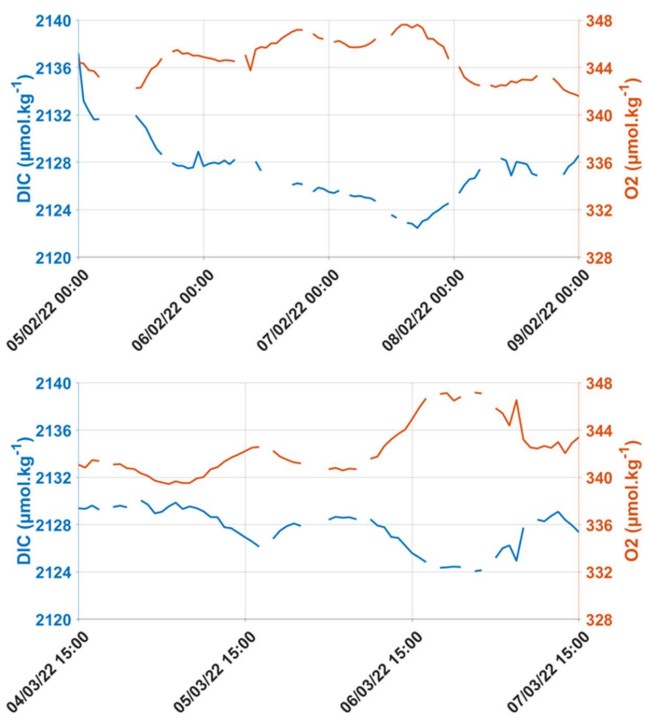

**Figure A1: DIC and O$_2$ (uncorrected) for periods during which NCP was estimated, in summer 2022.**

**Appendix B: Comparison between salinity products and CARIOCA SSS**

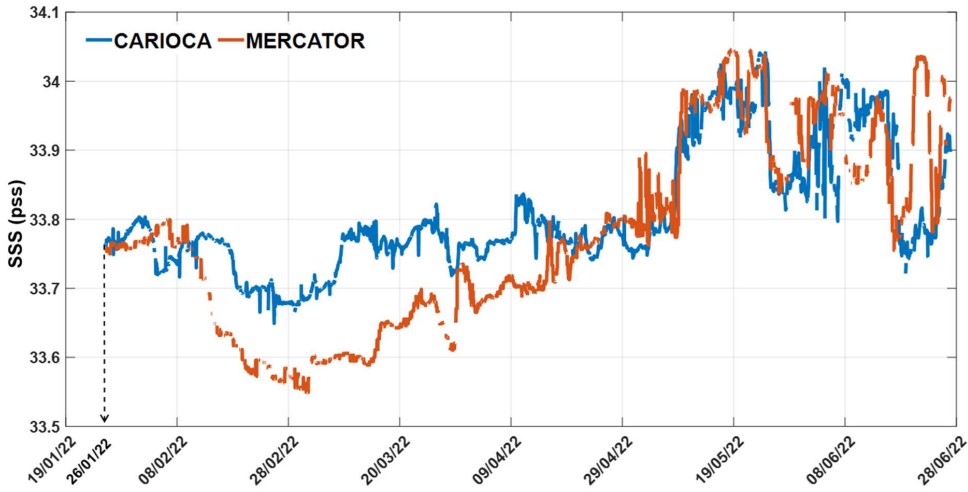

**Figure B1: Colocalization between CARIOCA hourly SSS, at 2 m, and Mercator's daily salinity, at 5 m, from the 26 January to the 27 June 2022.**

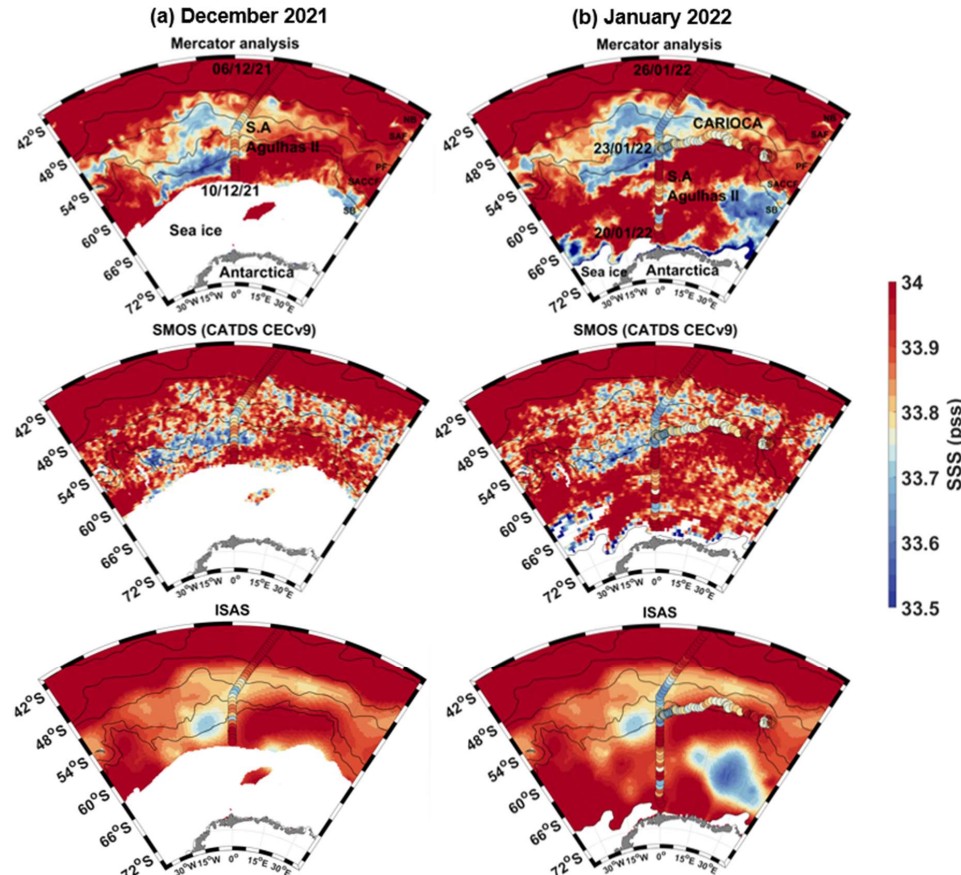

**Figure B2: Mercator salinity at 5 m, SMOS CATDS CECv9.0 SSS, and ISAS (MY) salinity at 5 m, with sea ice mask (0 % threshold of sea ice concentration) and fronts superimposed, (a) December 2021 mean SSS map, with *S.A Agulhas II* SSS from the 6-10 December 2021, (b) January 2022 mean SSS map, with CARIOCA SSS from the 26 January 2022 to the 27 June 2022, and *S.A Agulhas II* SSS from the 20-26 January 2022. SMOS SSS was adjusted by -0.1 pss to fit with CARIOCA and *S.A Agulhas II* SSS.**



**Appendix C: Origin of fresh water mass**

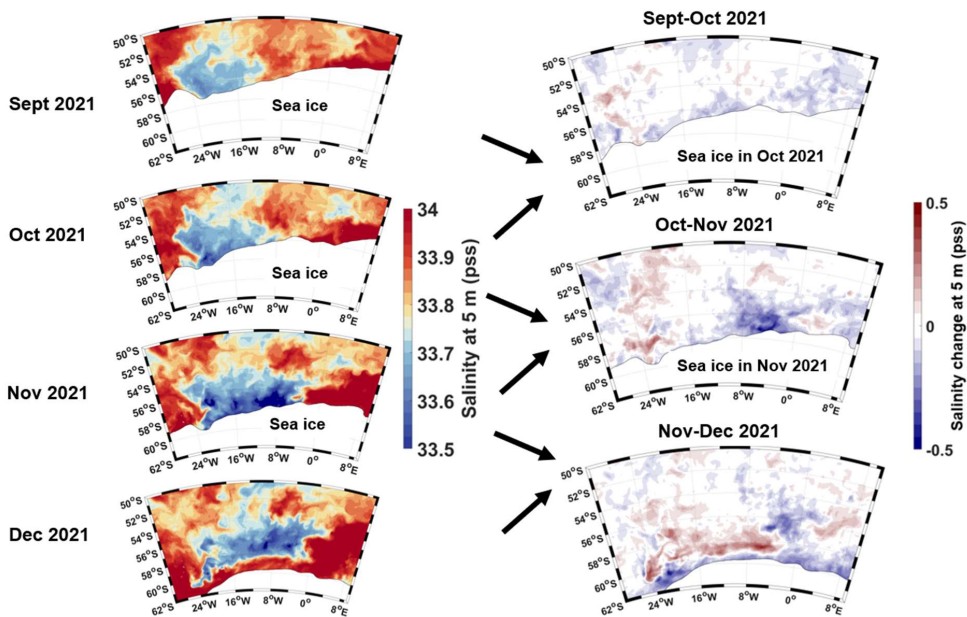


**Figure C1: (a) Mercator salinity at 5 m, with sea ice (10 % threshold of sea ice concentration, from OSI SAF) receding**
**from September to December 2021, (b) corresponding salinity change maps.**
















**Appendix D: Comparison between summers 2022 and 2019**

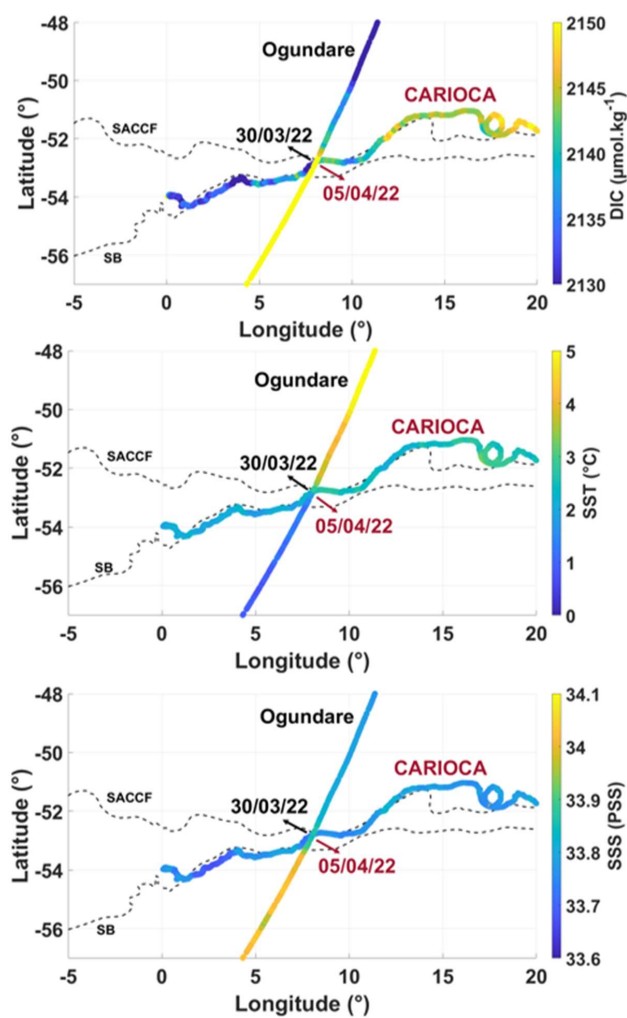


**Figure D1: DIC, SST and SSS of CARIOCA SO-CHIC (2022) and Ogundare (2019).**



**Appendix E: Origin of summer 2022 phytoplankton bloom**

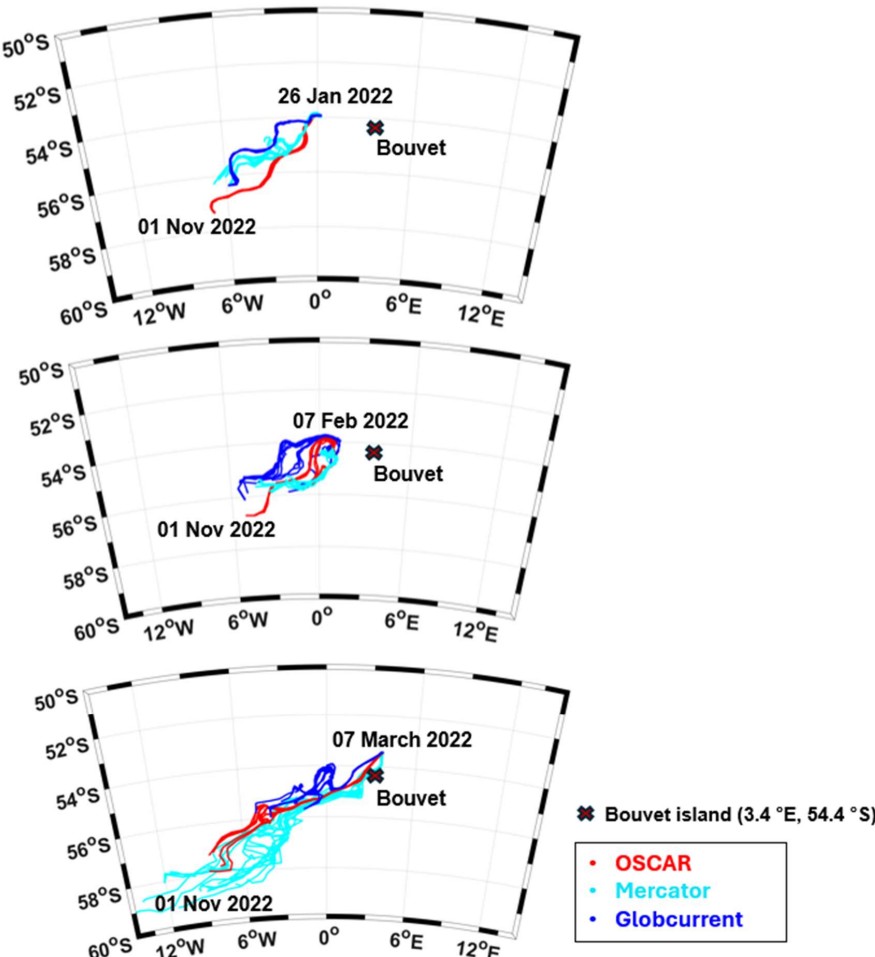


**Figure E1: Backward trajectories from January, February, and March 2022 to November 2021, using different current**
**products (Mercator analysis, OSCAR, Globcurrent).**









**Appendix F: Estimation of DIC and fCO$_2$ variations along the water mass trajectory**

Neglecting mixing and neglecting dfCO$_2$ due to salinity change,

the monthly changes of fCO$_2$ are estimated as in Merlivat et al. (2015):

$\Delta fCO_2 = \Delta fCO_{2SST} + (\delta fCO2/\partial DIC)\ \Delta DIC$        (F1)

*With*

$dfCO_2/dt = (\partial fCO_2/\partial T)\ dT/dt + (\partial fCO_2/\partial DIC)\ dDIC/dt$        (F2)

$\partial fCO_2/\partial T = (0.0423\ °C^{-1})\ fCO_2$        (F3)

$\partial fCO_2/\partial DIC = R(fCO_2/DIC)$, where R is the Revelle factor        (F4)

Neglecting mixing: $\Delta DIC = \Delta DIC_{(bio)} + \Delta DIC_{(air-sea)}$        (F5)

Where $\Delta DIC_{(bio)}$ is the contribution from biological production, and $\Delta DIC_{(air-sea)}$ is the contribution from the air-sea flux.

$\Delta DIC_{(air-sea)} = F / (\rho \ast MLD)$, where $\rho$ is the seawater density and F is the air-sea flux (Eq.(1))        (F6)

We assume an exchange coefficient, K, of 0.1/12 mol m$^{-2}$ month$^{-1}$ µatm$^{-1}$ (Boutin et al., 2009), and a Revelle factor, R, of 16 (Sabine et al., 2004).

According to (McClish and Bushinsky, 2023) the highest rates of daily net community production occur during active sea ice retreat and the bloom net community production is higher in case of early sea ice retreat. Given that the water mass reaching the CARIOCA subpolar region remained in the bloom near sea ice edge from September to November 2021, i.e. over the whole bloom period, it seems reasonable to consider that the DIC of this water mass was affected by a bloom NCP typical of early ice retreat, of 3 molC m$^{-2}$ bloom$^{-1}$ (Figure 4 of McClish and Bushinsky, 2023). The MLD measured by the glider in summer 2022 is around 60 m. Assuming the same MLD depth in spring 2021, we estimate that the $\Delta DIC_{(bio)}$ due to the bloom near sea ice edge in spring 2021 was -50 µmol kg$^{-1}$. Assuming a DIC concentration before melting around 2200 µmol kg$^{-1}$ (Briggs et al., 2018) and fCO$_2$ initially in equilibrium with the atmosphere (fCO$_2 \sim$ 400 µatm), the decrease in fCO$_2$ in November due to bloom NCP is estimated to be -160 µatm (November fCO$_2 \sim$240 µatm).

We then assume that, at first order, the fCO$_2$ change along the water mass trajectory is only affected by air-sea exchange and temperature change (i.e. we neglect biological production along the water mass trajectory and mixing with subsurface water, which seems at first order reasonable given the relatively low Chl-a along the water mass trajectory than in the vicinity of the sea ice edge, and given that the salinity anomaly persisted along the water mass trajectory until January 2022 respectively).

*From November 2021 to December 2021*, $\Delta DIC_{(air-sea)} = 22$ µmol kg$^{-1}$ and the $\partial fCO_{2(air-sea)}$ is +42 µatm. After the air-sea flux correction, we obtain a DIC concentration of about 2172 µmol kg$^{-1}$, and an fCO$_2$ of 282 µatm. The waters from sea ice melt are at -1.8 °C (seawater's freezing point). According to Remote Sensing System (REMSS, C.f. Data sets) and to backward trajectories, the SST along the water mass trajectory was -0.5 °C at the end of November 2021.Considering a heating of 1.3 °C ($\partial T$ = +1.3 °C) during the month of November, the temperature effect leads to an increase in fCO$_2$ of +15 µatm. At the end of December 2021, the fCO$_2$ is therefore 297 µatm.

*In January 2022*, $\Delta DIC_{(air-sea)}$ is +14 µmol kg$^{-1}$, and the $\partial fCO_{2(air-sea)}$ is +31 µatm. After the air-sea flux correction the fCO$_2$ is 328 µatm and the DIC is 2186 µmol kg$^{-1}$. According to REMSS and to backward trajectories, $\partial T$ is +1.0 °C, the $\partial fCO_{2(SST)}$ is 14 µatm. At the end of January 2021, the fCO$_2$ estimate is therefore 342 µatm.



**Data Availability**

All data used in this study are freely available and downloadable from the following websites. CARIOCA dataset is available on SEANOE at https://doi.org/10.17882/100800. The Seaglider (675) dataset is available online at https://doi.org/10.5281/zenodo.11059426. The *S.A Agulhas II* SO-CHIC 2022 cruise TSG dataset is available on SEANOE at https://doi.org/10.17882/100905. The S.A.*Agulhas II* SO-CHIC 2022 cruise, CTD data is available on SEANOE at https://doi.org/10.17882/95314. ERA5 atmospheric pressure and wind speed hourly data on single levels from 1940 to present is available on Copernicus Climate Change Service Climate Data Store , at https://doi.org/10.24381/cds.adbb2d47. The « Global Ocean Physics Reanalysis » and the « Global Ocean Physics Analysis and Forecast » datasets are available on Copernicus Marine at https://doi.org/10.48670/moi-00021 and https://doi.org/10.48670/moi-00016. The SMOS CATDS CECv9 salinity dataset is available online at https://doi.org/10.17882/52804#102161. The ISAS MultiYear salinity dataset is available on the data store of Copernicus Marine, at https://doi.org/10.17882/46219. The OSCAR dataset is available at https://doi.org/10.5067/OSCAR-03D01. The Globcurrent dataset is available at https://doi.org/10.48670/mds-00327. The OCI-CCI chlorophyll-a dataset is available online at http://www.oceancolour.org. The MW OI SST dataset produced by Remote Sensing Systems is available online at www.remss.com. The datasets "Global Sea Ice Concentration Climate Data Record v3.0 - Multimission, EUMETSAT SAF on Ocean and Sea Ice" and "Global Sea Ice Concentration Interim Climate Data Record Release 3 - DMSP, EUMETSAT SAF on Ocean and Sea Ice" are available online at http://doi.org/10.15770/EUM_SAF_OSI_0013 and http://doi.org/10.15770/EUM_SAF_OSI_0014. The SPASSO package containing the LAMTA software used for the lagrangian analysis can be downloaded at https://www.swot-adac.org/resources/spasso/

**Supplement**

The supplement related to this article is joined as a separate file to this submission.

**Author contributions**

J.Boutin oversaw the study, helped in the data processing, in the estimations, in the analysis and interpretation of the results and in the conception of the paper. S.Swart and M.du Plessis provided the Seaglider (675) dataset. They also provided useful input concerning the Seaglider data processing and interpretation of the results. L.Merlivat provided useful inputs and insights during the progress of this study, suggesting various hypotheses and helping us investigate them. L.Beaumont oversaw the verification of the calibration of CARIOCA sensors. A.Lourenco helped in the deployment of the CARIOCA buoy during the SO-CHIC 2022 cruise. Their involvement during the COVID period enabled us to collect data vital to this study. F.d'Ovidio and L.Rousselet provided the python code used for the Lagrangian analysis, which is the basis of our study and without which we couldn't have proved the origin of the phytoplankton bloom. They helped with discussions and modifications related to this code, for our specific study case. B.Ward provided the TSG data of the S.A.*Agulhas II* ship (SO-CHIC 2022). J.B.Sallée coordinated the SO-CHIC project, and provided part of the financial support. All co-authors discussed, reviewed, and edited the paper.



**Competing interests**
The contact author has declared that none of the authors has any competing interests.
**Acknowledgements**
The authors would like to thank Dimitry Khvorostyanov (dimitry.khvorostyanov@locean.ipsl.fr) and Alain
Laupin-Vinatier (alain.laupin-vinatier@locean.ipsl.fr) for the automation of CARIOCA data processing chain in
real time. We also thank S.Nicholson and M.Vancoppenolle for helpful discussions.
**Financial support**
This project has been partly funded by the SO-CHIC programme. L. Rousselet is supported by the CNES TOSCA
Bioswot-AdAC project.

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
