# Peer review of "Anomalous summertime CO2 sink in the subpolar Southern"

_EGUsphere, 2024_

## Author Comment (AC1)

**Review 1:**

**General**

We thank the reviewer for taking the time to review our paper and for constructive and helpful feedback. We have considered each comment carefully and reply to them below.

This is an interesting work with significance for research communities interested and working on similar topics. It shows the importance of in-situ observations and the synergies between different platforms to better understand the spatio-temporal dynamics of biogeochemical parameters coupled to physical phenomena. The hypothesis on what is driving the CO2 sink anomaly is sound (i.e. dissolved iron), however the results from this study only make an indirect connection and do not provide a way to quantify this.

We agree with the reviewer. And our hypothesis that dissolved iron is responsible for a bloom near sea ice edge, as suggested by some studies (Fitch and Moore, 2007), is not at all demonstrated by our study. Iron released during ice melt (Lancelot et al., 2009, Lannuzel et al., 2016, Person et al., 2021) probably contributed to the development of the phytoplankton bloom at the ice edge in spring 2021, but other factors as well, such as stratification due to fresher waters released during melting, light availability, grazing pressure and phytoplankton community composition (Smith and Comiso, 2008, Briggs et al., 2018, Mc Clish and Bushinsky., 2023) should also be considered. We have therefore modified this statement, accordingly, putting emphasis on the early sea ice retreat (of spring 2021, as referred in Wang and al., 2022) instead of the dissolved iron only.

The arguments and references in the discussion section provide a justification but not strong enough to have an explicit statement as the one mentioned in the abstract (line 24).

Yes, the reviewer is right, we have removed that statement. (See the comment above.)

The methodological approach and the subsequent analysis are robust to constrain air – sea CO2 fluxes, considering the inherent limitations that in situ- technology and the deployment area impose. The use of satellite data is coherent and complements well that in-situ observations. One surprising point is that SOCAT, GLODAP, BGC ARGO/SOCCOM data (exception: the Briggs et al., 2018 paper) are not used or explored to identify whether they can increase the data density and/or comparisons with previous years. If such data are not relevant due to spatio-temporal differences, it will be useful to mention it. To an extent this might strengthen the relevance of the dataset and work as the dataset will be even more valuable.

We chose to show comparisons to the year 2019, as this was the year in which the most data were available at the same position and same dates as the CARIOCA. We used the waveglider data (Nicholson et al., 2022) and did a comparison with data from Ogundare (Fig. D1) (Ogundare et al., 2021).

However, the reviewer is right, it would be useful to increase the comparisons using individual SOCAT datasets. We have replaced section 3.4 of the paper, by a comparison between the CARIOCA and the previous SOCAT campaigns. We have added a histogram of the dfCO$_2$, around 0 °E, 54 °S in summer, for the previous years, using all the available SOCAT data from 2000 to 2022. Section 3.4, showing the comparison between summers 2022 and 2019, has been

moved to the appendix. (See the answer to the comment below, on section 3.4 of the results, to get a detailed description.)

Concerning GLODAP, there were no data in summer 2022 around 0 °E, 54 °S. However, the climatology profiles (GLODAP mapped climatologies v2 2016b) of DIC and $O_2$ are coherent with the hypothesis of DIC increase and $O_2$ decrease, during the mixing events identified in February and March 2022. (See the reply to the comment at line 223, in the results section below).

There were very few BGC ARGO floats around the CARIOCA during its whole trajectory and most of them were too far away to be of any relevance. In summer 2022, there was only one float spatially and temporally close to the CARIOCA, on the 23rd of January 2022. This BGC ARGO float only measured T, S and fluorescence, and didn't bring any additional information to our study as we already had vertical profiles from the glider dataset, which are even more accurate since the glider followed the CARIOCA and stayed very close to the buoy for one month and a half. Moreover, the only BGC ARGO floats with pH measurements (which would have enabled us to approximate the $pCO_2$) around the CARIOCA's path, didn't sample the same water masses (different SST), as they were farther away. There was however one BGC ARGO float with oxygen measurements, close to the CARIOCA and which sampled the same water mass (same SST and SSS), on the 23 October 2022 (see the comment in the methods section below).

On that note the relevance and impact of this work is evident, especially if one considers that it's in a severely under-sampled areas and with a large contribution to earth's climate.

**Specific comments on sections**

**Introduction**

This section is concise and comprehensive. The reference coverage is ok but some additional references like, Sutton, A. et al., 2021 (https://doi.org/10.1029/2020GL091748), Landschützer, P. et al., 2015 (https://doi.org/10.1126/science.aab2620), Sarmiento, J. et al., (https://doi.org/10.1016/j.pocean.2023.103130) seem relevant.

The reviewer is totally right, we have added a few lines to the introduction, citing these references.

Line 52: For additional simplicity it might be useful. The references are adequate, but it will be helpful for the reader if values for interdecadal and interannual variability are mentioned.

We have added the following text to Line 55: Present-day sampling by $fCO_2$ products may overestimate the decadal variability of the Southern Ocean carbon sink by 50 %–130 % due to data sparsity (Hauck et al., 2023). When derived from the $pCO_2$ product, mean decadal amplitude is 6.31 Tmol $yr^{-1}$ and mean interannual amplitude is 1.68 Tmol $yr^{-1}$. On the other hand, when derived from the GOBMs, both mean decadal amplitude and mean interannual amplitude is 2.1 Tmol $yr^{-1}$ (Mayot et al., 2023).

Line 74 – 75: Please add some references

Here are the references: Ardyna et al, 2019, Boyd et al., 2007; Blain et al., 2008.

According to Merlivat et al., 2015, in introduction section, at p. 2: « In high-nutrient, low-chlorophyll (HNLC) regions, including the Southern Ocean, more than 2 decades of intense research have confirmed that increasing iron supply stimulates primary production (Boyd et al., 2007; Blain et al., 2008). »

Line 77: The point is not visible in Fig 1.

We have modified this figure accordingly.

**Methods**

Overall, the section is well presented. The lack of calibrations procedures for the optode is a short-fall. The authors do explain how this is addressed but there's no assessment of whether this assumption is adequate for this work. The 8 uM correction is high and there is a question whether this is drifting with time. The fact that O2 is used as a diagnostic and showcasing trends, makes this shortfall less important, however there might be an impact on the O2-O2sat calculation.

Without any correction (i.e. considering only the factory calibration of the optode), we observed that $O_2$-$O_2$sat was negative during some periods in summer 2022 while the CARIOCA buoy and the glider were detecting Chl-a and biological activity (as shown by measured fluorescence and by diurnal cycles of $O_2$-$O_2$ sat and DIC in opposition phase on the 5-7 February 2022). Unfortunately, we were not able to correct for the optode calibration using other measurements because there were no oxygen CTD measurements corresponding to CARIOCA measurements, and the glider's oxygen data was also uncorrected and could not be used as reference. Moreover, there were no BGC ARGO float with oxygen measurements near the CARIOCA, in summer 2022. We chose an arbitrary correction of + 8 µmol kg$^{-1}$ to ensure positivity of $O_2$-$O_2$sat in summer 2022. This correction has no impact on the trend of $O_2$-$O_2$ sat, it only brings the values +8 µmol kg$^{-1}$ higher. Without any way to correct the data, the correction of + 8 µmol kg$^{-1}$ was therefore most logical choice for the period shown in our paper (26 January to the 27 June 2022). Since we are looking at the trend, this does not change the interpretation of our results.

On the 23 October 2022, there was a BGC ARGO float with oxygen measurements, close to the CARIOCA, and which sampled the same water mass (same SST and SSS). This BGC ARGO measured oxygen values of 342 µmol kg$^{-1}$, at 78.1 °E, 53 °S, while the CARIOCA measured a mean oxygen value of 321.2 µmol kg$^{-1}$ on the same date, at around 77.6 °E, 54.4 °S. Considering this, an even higher correction, of + 20.8 µmol kg$^{-1}$ (instead of +8 µmol kg$^{-1}$) should be applied to the CARIOCA oxygen measurements as from the 23 October 2022. However, we cannot exclude that the correction evolved between beginning 2022 and spring 2022. In particular, we observe a negative jump of approximately 10 µmol kg$^{-1}$ in $O_2$-$O_2$sat on the 17/09/2022. Adding this jump to the correction of 8 µmol kg$^{-1}$ at the beginning of the time series would make it close to the BGC ARGO float observation.

[Figure]

(This figure is shown here for reference but has not been added to the paper.)

We kept a correction of + 8 µmol kg$^{-1}$ for the whole time series and specified in the legend of figure S2 that there might be an underestimation of about 10 µmol kg$^{-1}$ for $O_2$-$O_{2sat}$, as from the 17/09/2022 (Cf. our response to reviewer 2: we have added a figure similar to figure 2 of the paper but with the CARIOCA whole time series, to the supplementary material).

We have also modified line 93: According to the manufacturer, the accuracy of the $O_2$ measurements is ~ 16 µmol kg$^{-1}$ (5 %).

Why is the oxygen flux (FO2) mentioned (see comment in results as well) since it's not used later in the results or analysis?

$FO_2$ is used when estimating the $NCP_{O2}$, following the same methodology as in Merlivat et al., 2015 (we used the same methodology as in Merlivat et al., 2015, but we used the mixing layer depth to estimate the NCP, as in Merlivat et al., 2022). We have added this to the line 125 of the paper, explaining how the $FO_2$ is used.

Line 91-92: Please mention the model, type of sensors and how good they performed.

-The seawater dissolved $pCO_2$ sensor is a spectrophotometer (manufactured by NKE-France), using thymol blue dye and doing measurements at three wavelengths (810, 434 and 596 nm), with an accuracy of ± 3 µatm (Copin-Montégut et al., 2004, Boutin et al., 2008).

-Conductivity (accuracy: ± 0.003 mS cm$^{-1}$) via SBE37 SI probe, from which SSS was derived (Sea-Bird salinometer, model Microcat SBE 37 SI).

-Seabird SST sensor (Accuracy: ± 0.002°C for temperatures between -5 and 35°C).

- Dissolved Oxygen via (Aanderaa) Optode 4835, with an accuracy of 16 µmol kg$^{-1}$ (5%).

-Fluorescence (Chlorophyll) via Wetlabs fluorometer

Also, refer to Copin-Montégut et al., 2004 and to Boutin et al., 2008, for a more detailed description of how CARIOCA sensors generally work. A small table summarizing this has been added to the supplementary material.

Line 93: Is the unit umol l$^{-1}$ or umol Kg$^{-1}$?

The optode measurement for oxygen is in µmol L$^{-1}$, then I converted it to µmol kg$^{-1}$, for consistency throughout the paper.

Line 127: typo

Thank you, we have corrected this.

Line 133: How close is "nearest"? Not clear whether this is 20 km or less.

It is 20 km or less, between the 31$^{st}$ of January and the 10$^{th}$ of March 2022. We have added this to the paper at line 133. Below is the graph showing the distance between the CARIOCA and glider. We will not add this figure to the paper, but we show it here for reference.

[Figure]

**Results**

Surprised that flux data for both CO2 and O2 are not presented at all (even in the appendix). The authors do make a valid point of the use of DIC and NCP fluxes, however it's a bit confusing to mention sinks and fluxes in the title, explain how they are calculated and mention them throughout the manuscript and don't include a single graph.

The reviewer is right, we have included the $CO_2$ flux in the figure 2 (a).

[Figure]

Figure 2: CARIOCA time series from the 26 January 2022 to the 27 June 2022: (a) Atmospheric and surface ocean $fCO_2$ ($fCO_{2atm}$ and $fCO_2$), and daily mean of $CO_2$ flux, (b) DIC and wind speed, (c) SST and SSS, (d) Chl-a and $O_2$-$O_{2sat}$, with the period during which the glider followed the buoy (31 January to 10 March 2022) indicated by dotted lines.

Line 223: "…probably entraining waters rich in DIC…". Can GLODAP provide more quantitative information on this and enhance the XLD analysis?

There were no GLODAP data in summer 2022 around 0 °E, 54 °S. However, we can make an estimation using GLODAP climatological profiles of DIC and $O_2$ (using mapped climatologies v2 2016b, Lauvset et al., 2016). For instance, we plotted vertical profiles of DIC and $O_2$ using climatological GLODAP data at 0.5 °E, 54.5 °S, corresponding approximately to the location of the CARIOCA between the 7th and 9th of February and at 4.5 °E, 53.5 °S, corresponding to the event between the 14th and 17th of March.

In the paper, at lines 222-223, we mention a mixing event (after the 15th of March) during which DIC was likely brought to the surface from the subsurface layer. The GLODAP climatological profile (mentioned above) reinforces this hypothesis, with DIC increasing sharply from 50 m depth. For the mixing event between the 7th and 9th of February, using GLODAP climatology profiles at 0.5 °E, 54.5 °S, for an XLD deepening from 40 m to 80 m, the GLODAP climatological profiles show a DIC increase of about 10 µmol kg$^{-1}$ and $O_2$ decrease of about 7.5 µmol kg$^{-1}$, which is relatively coherent with the changes observed by the CARIOCA (Cf. figure (a) below and figure 4 of the paper).

These profiles also show that DIC varies more rapidly than $O_2$ at depth, explaining the lower diminution of $O_2$ compared to the DIC increase, during the mixing event in March. According to the figure (b) below, a DIC increase of about 15 µmol kg$^{-1}$, and an $O_2$ decrease of < 5 µmol kg$^{-1}$, can be expected due to an XLD deepening from 60 m to 90 m, which is relatively coherent with the DIC increase and $O_2$ decrease measured by the CARIOCA, between the 14th and 17th of March (Cf. Figure 4 of the paper).

[Figure]

Figure: Vertical profiles of DIC and oxygen (using GLODAP mapped climatologies v2 2016b) at (a) 0.5 °E et 54.5 °S corresponding to CARIOCA's location between the7[th] and 9[th] of February 2022 and at (b) 4.5 °E et 53.5 °S, corresponding to CARIOCA's location between the 14[th] and 17[th] of March 2022.

In the paper we have added that the DIC and $O_2$ changes we observe are consistent with the orders of magnitude that could be derived using GLODAP climatological DIC and $O_2$ profiles, considering the change in XLD.

Section 3.4: The comparison is valid, yet difficult to provide strong conclusions considering that it's a comparison against only one season 2 years before. Maybe SOCAT data can provide a bit more coverage.

We chose to go into detail about the comparison of the CARIOCA data with the waveglider (Nicholson et al, 2019), because both instruments were around 0 °E, 54 °S, at the same dates, enabling us to do a comparison of T, S, $fCO_2$, MLD and DIC between summers 2022 and 2019. We feel that this provides an ideal case study to illustrate the high-frequency processes that lead to interannual variability of variables that are difficult to obtain. We also mentioned Ogundare (2019) dataset, showing the DIC variations north and south of the SACCF in 2019, and Takahashi's climatology, for more consistency. Nonetheless, the reviewer is right, we have added to the paper a histogram comparing $dfCO_2$ for all SOCAT data available in January and February, between 2000 and 2022, near 0 °E, 54 °S.

[Figure]

Figure: Histogram of dfCO$_2$ in January and February, estimated using all SOCAT data available between 0 °E – 3.5 °E and 53 °S – 55 °S, from 2000 to 2022. The summer 2019 data is from the waveglider (Nicholson et al., 2022).

The SOCAT measured only act to corroborate our findings. There was outgassing in summers 2008 and 2010 and in 2019 the fCO$_2$ of the ocean was close to equilibrium with the atmosphere. This was observed in Nicholson et al. (2019). During these years, there was no local bloom, with backward trajectories demonstrating that the waters reaching 0 °E, 54 °S did not come from a spring bloom near the ice edge either. On the other hand, in 2014, 2015 and 2020, a small sink was observed in summer (Jan - Feb) even though there was no local bloom. The waters reaching the region these years, as identified from backward trajectories, came from a bloom near sea ice edge in November (except in February 2014, when the waters came from a bloom occurring in December). Similarly, in 2016, the waters might have come from a November bloom near the ice edge, but due to missing data (on CCI Chl-a maps), we cannot be sure. However, these spring blooms were less intense than the one in 2021 (our study case). (Refer to Figure 9 of the paper, comparing the intensity of Chl-a at both ends of the backward trajectories, in February and November, for all the years.) In 2009, when the fCO$_2$ of the ocean was close to equilibrium with the atmosphere, there was no local bloom, but the waters reaching the region came from a November bloom. However, it was a less intense bloom and there might have been mixing the following months, explaining the smaller dfCO$_2$ compared to the other years. Summer 2022 was the first summer with such a large negative dfCO$_2$ around 0 °E, 54 °S and demonstrates the importance of sea ice and freshwater advection for carbon variability in the Southern Ocean.

We have replaced section 3.4 of the paper with the figure above (histogram of SOCAT dfCO$_2$) and added a detailed description of the comparison between the CARIOCA and the previous SOCAT campaigns. Section 3.4 in the original manuscript – showing the comparison between summers 2022 and 2019 – has been moved to the appendix.

**Discussion**

Line 316: What's the definition of "massive"? Also related to the general comment in the results section, is the fact that there's no value with the term "unusually large CO2 sink".

Thank you, this was an oversight on our part. The CARIOCA $dfCO_2$ is on average about $-60$ µatm from January to March 2022, with $dfCO_2$ reaching values as high as $-90$ µatm, during periods of high local biological activity.

These values are close to the ones usually observed around Kerguelen (which is an area known for high biological activity) in summer and spring. We did a comparison (not shown in the paper) to the KEOPS2 (Kerguelen Ocean and Plateau compared Study expedition) campaign in November 2011, during which a CARIOCA drifter had been deployed east of Kerguelen Island (Merlivat et al, 2015). In spring 2011, $dfCO_2$ as low as -70 µatm were measured east of Kerguelen. We also found that NCP values estimated in summer 2022 east of Bouvet were, although smaller, close to the ones measured east of Kerguelen in spring 2011. Similarly, in November 2016, during the SOCLIM project (Pellichero et al., 2020) a CARIOCA sensor was used at a fixed mooring near Kerguelen and measured $dfCO_2$ as low as -90 µatm.

Moreover, when comparing to $pCO_2$ climatologies (Takahashi et al, 2012), the region observed by the CARIOCA in summer 2022 is usually in equilibrium with the atmosphere or is a much smaller sink. Also refer to the histogram in Section 3.4 above, showing the $dfCO_2$ values for every SOCAT data available in that region in summer, compared to CARIOCA $dfCO_2$.

Line 322: Again what's the definition of "more massive"?

We have replaced the term 'massive' by orders of magnitudes in the paper.

When we used the term massive, we were referring to values of Chl-a larger than 1 mg m$^{-3}$. From January to March 2022, the CARIOCA measured Chl-a concentrations between 0.8 and 1.2 mg m$^{-3}$, with values reaching as high as 1.4 mg m$^{-3}$ in March. In comparison to previous years, Chl-a concentrations in that area usually do not exceed 0.6 mg m$^{-3}$, even with the synergy of iron from Bouvet Island and from hydrothermal vents of seamounts (Sergi et al, 2020).

4.2. Interannual variation: It would have been useful to elaborate more on air-sea co2 fluxes variability.

We have added a reference to the figure 2 (b) - (d), of Gruber et al., 2019, which shows that climatologies of yearly air-sea $CO_2$ fluxes are close to 0 mol m$^{-2}$ in the region sampled by the CARIOCA (around 54 °S, east of 0° E). The CARIOCA $CO_2$ flux integrated over the studied period (26/01/2022 – 27/06/2022) is -1.88 mol m$^{-2}$, which, assuming negligible flux the rest of the year, is considerably greater than the climatological values in that region.

We have added a few lines describing the CARIOCA $CO_2$ flux in comparison to the ones shown in Gruber et al., 2019 to section 4.2 of the paper.

Moreover, as stated above (Cf. our reply to the comment on section 3.4), among all SOCAT campaigns, the CARIOCA in summer 2022 observes the largest negative $dfCO_2$.

1. Wondering whether it will be informative to show and attempt connections with sea ice retreat anomalies. The paper from Morioka, Y., et al., 2024 https://doi.org/10.1038/s43247-024-01783-z is very recent and wouldn't have been possible to include it but might be a source of inspiration(?)

Thank you very much for this valuable suggestion. We agree that there is a considerable interest in sea ice retreat anomalies, particularly since the notable decline in sea ice concentration since 2016 and what impact this might have on $CO_2$ fluxes in the Southern Ocean. While we do not include this in the current manuscript, this will be of interest for future work.

---

## Author Comment (AC2)

**Review 2:**

Review of "Anomalous Summertime CO2 sink in the subpolar Southern Ocean promoted by early 2021 sea ice retreat" by Kirtana Naëck et al.

This is an interesting study, that reveals the importance of sea ice melt or iron to support large phytoplanktonic blooms in the Southern Ocean, even in areas where enrichment or iron from land aeolian transport or hydrothermal vents could have been suspected at first sight. In addition, the authors put the observed bloom in perspective with earlier ice retreat, a feature that is predicted to repeat more frequently in the future. I am on the same line as the authors and think this study is valuable and deserves publication. I have no doubts about that.

Thanks for your positive comment!

Still, I have a few concerns.

The first one does not necessarily need to be addressed. SO-CHIC project proposed a very original combination of Carioca and glider surveys. I'm wondering if more emphasis could have been put on the physical processes. Gliders provide very valuable information about the physical processes underneath the buoy. The concepts of MLD vs XLD have been rapidly addressed in the results, but their impact was not discussed in depth elsewhere.

The study from Pellichero et al., 2020, has demonstrated that a decrease in wind speed causes an increase in stratification, as shown by a shoaling of the MLD, leading to an accumulation of Chl-a at the surface and a decrease in DIC due to increased carbon fixation by phytoplankton. We suggest the same phenomenon for our study case, for the biological events between the 05/02/2022 – 07/02/2022 and between the 04/03/2022 - 07/03/2022. Comparison between MLD and XLD also enables us to identify mixing with waters from the subsurface. (Refer to lines 222-226 of the paper).

The NCP was estimated using the XLD. We have corrected line 125 of the paper: "… following the same methodology as in Merlivat et al. (2015) but using the mixing layer depth as in Merlivat et al. (2022) and Pellichero et al. (2020) (see section 2.3).

We have corrected the title of Section 3.2 by: "NCP and impact of wind on XLD". Moreover, the biological event was between the 5[th] and the 7[th] of February, we have corrected the dotted green line on Figure 4 accordingly. We have also corrected the legend of the figure and the dates at line 213.

We have added the XLD on the glider profiles of figure 3:

[Figure]

Figure 3: Seaglider profile time series between the 31 January and the 10 March 2022 of (a) salinity, (b) Chl-a, with the MLD indicated by the black line and the XLD indicated by the purple line.

I have the feeling that the authors are doing a bit of cherry-picking, presenting only a part of the data set till 27 June 2022 or some periods where primary production strongly affects the DIC signal (first two weeks of February and first two weeks of march). Personally, I am not encouraging a selective or partial interpretation of the data, but I can understand that you want to deliver one message (the impact on primary production, away from the SIZ) and then not discuss every single aspect.

We decided to present only part of the data set, to focus on summer and autumn 2022 and explain these few months in more detail. The rest of the CARIOCA time series is part of another future study altogether.

We chose to go into more detail for the first two weeks of February and the first two weeks of March, because these are the only periods during which NCP could be estimated. Indeed, the NCP could only be estimated during periods of low wind speed, where we could suppose that there was no mixing with the subsurface and where diurnal cycles of DIC and $O_2$-$O_2$ sat in opposition could be observed. (On the right-hand side panel of Figure 4 (a), we have removed the MLD as from the 10th of March 2022, since after that date the glider stopped following the CARIOCA.)

Concerning the rest of the CARIOCA time series:

- There was an undersaturation of about -10 to -15 µatm in winter 2022 (with periods during which the $fCO_2$ of the ocean was in equilibrium with the atmosphere). This is surprising

since this region – the Polar Frontal Zone – is known to be at equilibrium with the atmosphere or should on the contrary show outgassing of $CO_2$ to the atmosphere (Gray et al. 2018). However, the region in which the CARIOCA was, has almost no winter data available. For the previous years, there were only a few SOCAT data in the winter season, between the Southern Boundary and SACCF, in the whole Southern Ocean, and even less in the Southern Indian Ocean.

Part of the undersaturation (about 5 µatm) observed by the CARIOCA can be explained by solubility changes, due to a negative SST anomaly in that zone in winter 2022. I am currently investigating further to explain the rest of this winter 2022 undersaturation.

-   Then, the CARIOCA observed another $CO_2$ sink (smaller than the summer sink), from November 2022 to February 2023, as it drifted south of Kerguelen Island, in a bloom coming from the Northern Plateau. Further east, as from February 2023, the CARIOCA $fCO_2$ increased back to values close to equilibrium with the atmosphere. Since the undersaturation wasn't observed the next winter (May-June 2023), this leads us to believe that the CARIOCA's $fCO_2$ measurements weren't biased.

We have added the figure below, showing the whole CARIOCA time series, to the supplementary material of the paper (the figures have been arranged to follow the order in which they are mentioned in the text).

[Figure]

Figure S2 : CARIOCA time series from the 26 January 2022 to the 24 June 2023: (a) Atmospheric and surface ocean $fCO_2$ ($fCO_{2atm}$ and $fCO_2$), and daily mean of $CO_2$ flux (b) DIC and wind speed, (c) SST and SSS, (d) Chl-a and $O_2$-$O_{2sat}$, with the period during which the glider followed the buoy (31 January to 10 March 2022) indicated by dotted lines. Since the CARIOCA observed a negative jump in the $O_2$ measurements on the 17/09/2022, there might be an underestimation of about 10 µmol kg$^{-1}$ for $O_2$-$O_{2sat}$, as from that date.

But then you should refrain from writing some sentences like "This suggests that biological activity was the dominant driver of the DIC seasonal variation "line 229, while you are looking into detail only some few weeks, and then during these few weeks, the DIC was actually increasing overall, not decreasing.

The reviewer is correct, the principal driver of the low DIC values in February and March 2022 is not the local biological activity. Although, the DIC slightly decreases during some periods of high local biological activity and low wind speed (when we could estimate the NCP), other processes, such as mixing and air-sea exchanges, are compensating this DIC decrease. The DIC concentrations are overall very low during the whole summer 2022, and we are suggesting that these low values are due to the fresh water mass, already poor in carbon, which came from the ice edge. We have removed line 229.

At the end, the incredible opportunity and amount of energy needed to deploy the glider in conjunction with the buoy was not really necessary. Results from the glider are not addressed in the discussion or in the conclusion.

The glider data were crucial to estimate the depth of the mixed layer (using density profiles, Cf. line 143 of the paper), and to analyse/separate the different contributions (biological activity, air-sea exchanges, mixing) on the DIC variations. Using the wind data, the MLD and XLD, we were able to identify mixing events with waters from the subsurface, and this could not have been done without the glider data. We have insisted on the glider's importance in the discussion.

The glider also enabled us to get in-situ vertical profiles of temperature, salinity and fluorescence, which provide a more comprehensive view, which wouldn't have been possible with only surface data. The glider vertical profiles showed that both the low salinities and high Chl-a concentrations were constrained by the mixed layer (Cf. lines 198 - 200). Moreover, since the CARIOCA Chl-a couldn't be calibrated with the ship CTD during its deployment (not enough points coinciding spatio-temporally), the glider was used to calibrate the CARIOCA Chl-a (and convert the arbitrary fluorescence units to Chl-a concentrations, cf. my comment below for line 141). This was referred to in the section 2.3 of the Materials and Methods, in the paper.

Second, there are more than 80 plots in the paper in three different places (in the text, in the 6 parts of the appendix and in the supplemental material). It's really a massive amount of information, and while it's a rigorous approach, to be honest, it confused me at some point. There is repeatedly the same information with different products (e.g. Figure B2) or the same data appears in different graphs (DIC changes and O2-O2 sat appear in figure 2, 4 and A1). The figures are not appearing in the same order that in the text (e.g. line 251 reads "Fig C1 and fig E1in appendix" – then why there is appendix D between C1 and E1).

There is a long discussion on Figure C1 that is a bit awkward for me and not needed. I mean, the presence of low salinity in the Southern Ocean due to sea ice melting is an information relatively straightforward and widely admitted.

The back trajectories are very useful, one distribution of salinity from remote sensing of reanalysis validated by the onboard thermosalinograph is largely enough for me, but personally, I don't need more proof of that. It is, of course, the authors' responsibility to add more figures, but keep in mind that at some point, the paper is becoming cumbersome with non-essential or repeated information. At some point, reading the results, I was starting to be confused, jumping from one figure to another one, in the main text, the appendix, or the supplemental, going back and forth. I would either remove some figures, like C1, or I would order them better, in only two places, in the order of appearance in the text, with careful references in the text to help the reader to follow your ideas.

The reviewer is right. The information given in figure A1 is indeed already available in figure 4. However, we kept figure 4, it is necessary to show a focus on the periods of high local biological activity, during which NCP was estimated. Figure 4 also shows a direct comparison of the MLD, XLD and wind to the DIC and $O_2$-$O_2$ sat, highlighting the diurnal cycles and the mixing events on synoptic time scales. Figure 4 has therefore been moved to the appendix, where it has replaced figure A1. We have simplified figure B2 by removing the ISAS comparison and we have removed figure C1.

We have removed figure E1 from the appendix and we have replaced figure 5 by the figure below:

[Figure]

Figure 5: (a) Backward trajectories from CARIOCA's location in January, February, and March 2022 to November 2021, using different current products (Mercator analysis, OSCAR, Globcurrent). (b) CCI Chl-a in November 2021, with the sea ice concentration at 10%, on the 1 November 2021, and backward trajectories superimposed.

We have reordered the figures in the appendix, according to the order of appearance in the text.

Minor comment

Line 79: It's up to you, but personally, I found the sentences "In the next section, the instruments deployed, the different data sets, and the methodology used will be described. There will then be a description of the results followed by a discussion " unnecessary. In each paper, that is roughly what we expect to find in that sequence. I would remove it therefore.

We have removed that part.

Line 84: "It was anchored at 15m". What does it mean? I presume that the buoy was NOT anchored, at least with a regular anchor, or it might have used a floating anchor, but then it should be precise.

Yes, it's a floating anchor, following the currents at 15 m depth, in a Lagrangian way. We have corrected this, at line 84, of the paper.

Line 90: Carioca buoys are formidable instruments, able to withstand the rigours of the Southern Ocean and provide precious data at mesoscale or synoptic time scales. Still, there has been a lot of discussion about the accuracy of pCO2 derived from drifters in the Southern Ocean (Long et al., 2021; Williams et al., 2017; Wu et al., 2022; Zhang et al., 2024), especially SOCCOM drifters. I acknowledge that SOCCOM and Carioca sensors are different, but that is still the same principle, and some potential biases are similar (measurement of pH instead of pCO2, assumption on alkalinity, and so on).

Contrary to SOCCOM floats, the CARIOCA sensor deduces the seawater $pCO_2$ using a spectrophotometer at three wavelengths and there is no assumption on alkalinity. Through a semi-permeable $CO_2$ membrane, seawater is brought in equilibrium with a dye solution, thymol blue, and the absorption coefficient of the dye is measured by the spectrophotometer (Copin-Montégut et al., 2004, as cited in our paper, Cf. lines 85 – 90). Indeed, while the SOCCOM floats measuring pH, rely on a hypothesis using an alkalinity-salinity relationship, the CARIOCA sensor measures the pH of a dye solution of which we already know the alkalinity (that is measured in the laboratory before deployment and that is independent of the sea water alkalinity). The $pCO_2$ calibration was done in the lab, using classical infrared $pCO_2$ measurements.

According to Copin-Montégut et al., 2004, in the methods section, p 172: "Carbon dioxide in a sea water sample equilibrates with a pH indicator solution across a gas permeable (silicon) membrane in an exchanger cell. The $pCO_2$ in sea water is calculated from pH and alkalinity of the dye solution at known temperature. The pH is measured using the light absorption properties of the thymol blue diluted in sea water with a constant alkalinity."

Also, refer to lines 88 – 90 of our paper: "The three wavelength measurements enable correction of any modification of the optical path or of the opacity of the optical cell (Copin-Montégut et al., 2004)."

When the error of ICOS measurements, with direct measurements of pCO2 using shower head equilibrator, CO2 CRD analysers, and regular calibration with standard gas is 2 µatm, I think that the absolute precision of 3 µatm claimed for Carioca buoy should be carefully assessed.

It corresponds to the expected accuracy of these instruments, given the accuracy of the laboratory calibration using IR instrument (Copin-Montegut et al. 2004). Previous checks performed at sea in the Southern Ocean confirmed this order of magnitude (Boutin et al. 2008, supporting information available on "https://aslopubs.onlinelibrary.wiley.com/action/downloadSupplement?doi=10.4319%2Flo.2008.53.5_part_2.2062&file=2062a1.pdf".)

I'm sure it has been, but a reference will be very welcome here. And do you mean precision or accuracy? The concept of absolute and relative precision, both in µatm is not clear to me. Is the

absolute precision the accuracy? Should the relative precision be in %? I'm sorry for my naïve questions.

The absolute precision mentioned here is the accuracy. Boutin et al., 2008, compared several CARIOCA drifters data in the Southern Ocean to ship measurements and concluded that the accuracy is 3 µatm and the precision is 1 µatm. We have added a reference to Boutin et al., 2008.

Line 102. Why stop on the 27 June 2022? It's somehow uncommon to present only a part of the data and not the full data set. Is there a scientific rationale for that? Was the rest of the data not interesting enough?

We have added a figure similar to figure 2, but showing the whole time series, to the supplementary materials. (Refer to my comment above, explaining the main points of the CARIOCA whole time series).

Line 128. Could you provide details on that instrument, brand, capability? I've tried to find some information on it, but it was not that easy to find some.

The lines 128-129 will be modified, adding these details: A Kongsberg Seaglider (SG675) was deployed alongside the CARIOCA buoy. A Seaglider is an autonomous underwater vehicle designed to fly through the water column from the sea surface to 1000 m depth following a sawtooth pattern, moving vertically and horizontally at nominal speeds of 0.1 m s$^{-1}$ and 0.3 m s$^{-1}$, respectively. Upon surfacing roughly every 6 hours, Seagliders communicate to base station via Iridium, thereby transferring data in near-real time. A key characteristic of Seagliders is their ability to be piloted from land, allowing researchers to control the direction of sampling. For this experiment, SG675 followed the CARIOCA buoy for a month and a half, from 31 January 2022 to 10 March 2022 (39 days) (Fig. 1), providing vertical profiles of temperature, salinity, oxygen and fluorescence of the upper 1000 m of the water column.

Line 135. This approach is interesting. Could you provide some details about the Savitzky-Golay filter that would be useful for others, like width and order?

We used a window width of 11 and an order of 2. For more details on this approach, refer to Gregor et al., 2019 and Swart et al., 2024 (cited at line 134 of the paper). On the figure below, the top panel shows the salinity obtained after smoothing the glider data with Savitzky-Golay, and the bottom panel shows the difference (in pss) from the original salinity.

[Figure]

Below are the salinity plots pre and post treatment (after applying all the corrections), with the last panel showing the difference between the two.

Physics Variable:
        Removing outliers with IQR * 2.5: 0 obs
        Removing spikes with rolling median (spike window=5)
        Removing horizontal outliers (fraction=0.2, multiplier=2.5)
        Smoothing with Savitzky-Golay filter (window=11, order=2)

[Figure]

These figures have not been added to the paper but are shown here for reference.

Line141. You're providing details on the calibration of the fluorimeter, but actually, it's calibrated against another fluorimeter (the one from the CTD). But then, how the fluorescence units are converted in chlorophyll (using built-in algorithms)? And why provide details for the sea glider and not from the Carioca buoy? It seems important for me to provide details on converting fluorescence to biomass.

There were no CARIOCA measurements coinciding with the CTD measurements, so we couldn't calibrate the CARIOCA directly using the CTD. On another hand, there were glider fluorescence measurements coincident with CTD fluorescence measurements and some of the latter were also coincident with CTD water samples from which the Chl-a was measured. Hence, since the glider followed the buoy, we calibrated the CARIOCA using the already calibrated glider measurements using CTD information.

Line 144. Remove one parenthesis after "de Boyer Montégut et al., 2004"

We have corrected this, thank you.

Line 144. "mixing" Should it be written "extreme layer depth" instead?

No, refer to Merlivat et al, BG 2022, paragraph 2.4:

"The mixing-layer depth, $Z_{mx}$, is the upper part of a mixed layer of uniform density where active turbulence occurs (Brainerd and Gregg, 1995)."

Merlivat, L., Hemming, M., Boutin, J., Antoine, D., Vellucci, V., Golbol, M., Lee, G. A., and Beaumont, L.: Physical mechanisms for biological carbon uptake during the onset of the spring phytoplankton bloom in the northwestern Mediterranean Sea (BOUSSOLE site), Biogeosciences, 19, 3911–3920, 2022.

Figure 4. Why did you split the figure and remove the second half of February? It looks like you're cherry-picking and presenting only part of the data set. This approach should be discouraged.

The NCP could be estimated only during periods where we can suppose that the mixing layer is isolated from the rest of the ocean, when the wind is low and there is no mixing with subsurface, and only when there are diurnal cycles of DIC and $O_2$-$O_2$ sat in opposition. Using the method from Merlivat et al, 2022, only these two periods of local biological activity could be identified. (We used the same methodology as in Merlivat et al. 2015, but instead of the MLD, we used the mixing layer depth, XLD, to estimate the NCP as in Merlivat et al. 2022. We have added a reference to Merlivat et al., 2022.)

(Also, as per our comment above: On the right-hand side panel of Figure 4 (a), we have removed the MLD as from the 10[th] of March 2022, since after that date the glider stopped following the CARIOCA.)

Line 229. "This suggests that biological activity was the 230 dominant driver of the DIC seasonal variation". There is something I don't quite understand. You put a lot of emphasis on the impact on biological production. I am not contesting that there was primary production, but,

when looking at the DIC, comparing the 01/02/2022 to 13/02/2022 or the 01/03/2022 to 17/03/2022, in both cases, the DIC is increasing overall, NOT decreasing. So there are other processes at work, mixing, air-sea exchange, and water mass change, that outweigh the decrease in DIC. I would not write, therefore, "the dominant driver". It is not for me.

The reviewer is correct. There were periods of local biological activity, during which the NCP could be estimated, but these were very short periods, spanning only a few days and were rapidly compensated by other events such as mixing. Outside the brief periods of local biological production and of mixing events, the DIC and the $fCO_2$ remained relatively stable at a low value in January and February. For instance, between the 11/02/2022 and the 03/03/2022, the DIC remained at a mean value of 2133 µmol kg$^{-1}$. It is very unlikely that the bloom observed locally, west and north-east of Bouvet Island, was what caused the very low values of $fCO_2$ observed. We put more emphasis on the fresh water mass, already poor in carbon, which came from the ice edge. We removed line 229 and we corrected the text accordingly.

Line 241 to line 245. Could you indicate the relevant figure? I am a bit lost among the 80 plots of the paper.

We added a reference to Figure 5.

Line 247. "The decrease in salinity started near the South Sandwich trench, around 25° W and 60° S, near the sea ice edge in September 2021". How can you see that? I mean, the data are blanked in the figure. I don't understand the discussion actually. The ice retreats, but in the west, there is low salinity, and in the East, there is high salinity.

Line 248: and 60° S, near the sea ice edge in September 2021, when the sea ice started to retreat

For simplicity, we removed these sentences at lines 246 – 249: "The formation of this fresh water mass can be seen, from September to December 2021, using salinity maps and sea ice data from OSI SAF (Fig. C1 in Appendix). The decrease in salinity started near the South Sandwich trench, around 25° W and 60° S, near the sea ice edge in September 2021, when the sea ice started to retreat. It continued to develop eastward near the sea ice edge until November 2021 (Fig. C1 in Appendix)."

Part 4.3. I'm not convinced by the rigour of the approach (assuming no mixing), but more importantly, by the interest of such computation subjected to caution, while it's not clear to me what this computation is bringing to the overall conclusion. It's not my decision, but I would remove that part.

The local summer bloom in which the CARIOCA entered in 2022, didn't seem to generate a decrease in $fCO_2$. Indeed, as the reviewer rightly pointed out, there was no overall decrease in DIC from January to March 2022. However, although there were sometimes slight increases in DIC due to other compensating events such as mixing, the $fCO_2$ and DIC values remained relatively low in summer 2022. Our hypothesis is that these very low $fCO_2$ and DIC values are not due to the local summer bloom in itself, but rather due to the fresh water mass, already poor in carbon, advected from the ice edge in November 2021 to the position of the CARIOCA in January 2022. Our computation reinforces our hypothesis, that the summer 2022 carbon sink was caused by early sea ice retreat. Indeed, for our estimations we use an NCP value typical of sea ice retreat starting as early as September, for the spring 2021 bloom (Cf. Line 497 of the paper). Concerning our assumption that there was no mixing; by cross-referencing different

salinity products (Mercator / Glorys and SMOS), we were able to determine that the SSS anomaly probably appeared at the same time as sea ice started retreating and then persisted several months at the surface and was advected to the CARIOCA's position in summer 2022 (not shown in the paper). Mercator's vertical salinity profiles (not shown in the paper), also confirmed the hypothesis of a surface fresh layer, which formed during ice melt and then stayed at the surface, and was advected north-east, for several months. For such a surface salinity anomaly to persist for so long, we therefore assume that there was probably very little mixing during that time. We added these details to the paper, putting more emphasis on the contribution of the waters advected from the ice edge.

Supplemental.

There is a poor definition of the figures when zooming in. It's difficult to see the details.

We have removed figures S3 and S4, since the information provided by these figures is already summarised by figure 9.

References

Long, M.C., Stephens, B.B., McKain, K., Sweeney, C., Keeling, R.F., Kort, E.A., Morgan, E.J., Bent, J.D., Chandra, N., Chevallier, F., Commane, R., Daube, B.C., Krummel, P.B., Loh, Z., Luijkx, I.T., Munro, D., Patra, P., Peters, W., Ramonet, M., Rödenbeck, C., Stavert, A., Tans, P., Wofsy, S.C., 2021. Strong Southern Ocean carbon uptake evident in airborne observations. Science 374, 1275–1280. https://doi.org/10.1126/science.abi4355

Williams, N.L., Juranek, L.W., Feely, R.A., Johnson, K.S., Sarmiento, J.L., Talley, L.D., Dickson, A.G., Gray, A.R., Wanninkhof, R., Russell, J.L., Riser, S.C., Takeshita, Y., 2017. Calculating surface ocean pCO2 from biogeochemical Argo floats equipped with pH: An uncertainty analysis. Global Biogeochemical Cycles 31, 591–604. https://doi.org/10.1002/2016GB005541

Wu, Y., Bakker, D.C.E., Achterberg, E.P., Silva, A.N., Pickup, D.D., Li, X., Hartman, S., Stappard, D., Qi, D., Tyrrell, T., 2022. Integrated analysis of carbon dioxide and oxygen concentrations as a quality control of ocean float data. Commun Earth Environ 3, 1–11. https://doi.org/10.1038/s43247-022-00421-w

Zhang, C., Wu, Y., Brown, P.J., Stappard, D., Silva, A.N., Tyrrell, T., 2024. Comparing float pCO2 profiles in the Southern Ocean to ship data reveals discrepancies (No. EGU24-3332). Presented at the EGU24, Copernicus Meetings. https://doi.org/10.5194/egusphere-egu24-3332

**Citation**: https://doi.org/10.5194/egusphere-2024-2668-RC2